# Mathematical Model for Regular and Irregular PV Arrays with Improved Calculation Speed

**Luz Adriana Trejos-Grisales** [1,*,†] and **Juan David Bastidas-Rodríguez** [2,†]
and **Carlos Andrés Ramos-Paja** [3,†]

1   Departamento de Electromecánica y Mecatrónica, Instituto Tecnológico Metropolitano,
    Medellín 050013, Colombia
2   Facultad de Ingeniería y Arquitectura, Universidad Nacional de Colombia, Manizales 170003, Colombia;
    jubastidasr@unal.edu.co
3   Facultad de Minas, Universidad Nacional de Colombia, Medellín 050013, Colombia; caramosp@unal.edu.co
*   Correspondence: adrianatrejos@itm.edu.co; Tel.: +57-4-4600727
†   These authors contributed equally to this work.

**Abstract:** Photovoltaic (PV) systems are usually developed by configuring the PV arrays with regular connection schemes, such as series-parallel, total cross-tied, bridge-linked, among others. Such a strategy is aimed at increasing the power that is generated by the PV system under partial shading conditions, since the power production changes depending on the connection scheme. Moreover, irregular and non-common connection schemes could provide higher power production for irregular (but realistic) shading conditions that aere caused by threes or other objects. However, there are few mathematical models that are able to predict the power production of different configurations and reproduce the behavior of both regular and irregular PV arrays. Those general array models are slow due to the large amount of computations that are needed to find the PV current for a given PV voltage. Therefore, this paper proposes a general mathematical model to predict the power production of regular and irregular PV arrays, which provides a faster calculation in comparison with the general models that were reported in the literature, but without reducing the prediction accuracy. The proposed modeling approach is based on detecting the inflection points that are caused by the bypass diodes activation, which enables to narrow the range in which the modules voltages are searched, thus reducing the calculation time. Therefore, this fast model is useful in designing the fixed connections of PV arrays that are subjected to shading conditions, in order to reconfigure the PV array in real-time, depending on the shading pattern, among other applications. The proposed solution is validated by comparing the results with another general model and with a circuital implementation of the PV system.

**Keywords:** photovoltaic array; general model; irregular configuration; partial shading; fast calculation

## 1. Introduction

The recent world challenges that are related to environmental and energy subjects have lead photovoltaic (PV) systems to become a competitive option for electricity generation. In recent years, this technology has gained popularity, which is evidenced in the 627 GW installed in 2019 [1] and in some reports that predict a growth of 46% by 2023 [2]. Therefore, it is important to develop procedures for predicting the behavior of such systems under a given operating condition. In this way, it will be

possible to improve planning and controlling strategies, which in turn will increase the reliability of PV plants. Mathematical modeling is a useful tool for analyzing the electrical relationships between voltages and currents in a PV array; it makes obtaining the current vs. voltage (I-V) and power vs. voltage (P-V) curves possible, which are needed for performing different kinds of studies on a PV array, such as energy estimation, degradation, and failure analysis, among others.

A typical PV array is formed by connecting multiple modules in series in order to form strings and multiple strings in parallel to form the array (i.e., Series-Parallel configuration). Each module can be represented by an equivalent circuit; therefore, the model of a PV array is obtained by analyzing the array equivalent circuit obtained by the interconnection of the modules. If all the modules are exactly the same and operate under the same irradiance and temperature conditions, all of the arrays can modeled by scaling the model of one module. Nonetheless, in real operating conditions the arrays are subjected to partial shading and the modules are different due to aging and manufacturing tolerances [3].

Concerning the PV module model, most of the works that are devoted to modeling PV arrays are based on the single-diode model. Other works, such as [4], adopt the two-diode model in order to improve the accuracy at low irradiance levels, but the computational burden increases significantly, because the two diode model has more parameters and greater non-linear elements. An interesting combination single-diode and double-diode models is proposed in [5]. In this work, the authors use a machine learning-based technique to calculate the power of PV arrays using single and double-diode module models. Nevertheless, this solution does not apply to any configuration and it does not consider partial shading or mismatching operating conditions.

PV arrays can be connected in different configurations, with the most common being Series-Parallel (SP), Total Cross-Tied (TCT), Bridge-Linked (BL), and Honey-Comb (HC) [6]. Other configurations are the result of a mix between those connection schemes, which can be considered as hybrid configurations [7]. Every configuration has particular characteristics imposing a different behavior to the PV array, such a situation can be exploited for improving the array power production under a particular operating condition. This requires identifying the configuration imposing the maximum power generation.

Different modeling procedures have been reported in the literature. Some of them only apply for SP [4,8–10] or TCT arrays [11,12]; while others [7,13] perform an independent analysis for SP, TCT, BL, and HC arrays, but they do not propose systematical procedure to model arrays in any configuration. Only in [14,15] fo the authors propose modeling procedures for arrays in any configuration. However, those works are focused on the generation of a system of nonlinear equations in order to model an array and the solution of the models requires high calculation times.

One technique that was used to improve the calculation speed in PV models was introduced in [10], named inflection points. Such work provides a modeling procedure in order to obtain the I-V and P-V curves based on the ideal single-diode model, but only for SP arrays. The inflection point concept is related to the operating conditions, in which the I-V curve changes its monotonic behavior due to the activation (or deactivation) of the bypass diodes, which occurs when the array operates under partial shading conditions. The inflections points allow for defining voltages ranges, in which, due to the activation of bypass diodes, some PV modules will be short-circuited; hence, those modules could be removed from the model. It means that the non-linear equations system, which represents the behavior of the array, will be dynamically modified, depending on the number of the active PV modules. Such a characteristic allows for increasing the calculation speed of the PV module, since less PV modules must be taken into account.

The inflection points concept was also used in [9] in order to reduce the calculation time for solving the model of SP arrays. In this paper, the authors analyze each string independently and uses the single-diode model to represent each module of the array. Subsequently, each string is modeled by a nonlinear equation where the string current is the unknown variable. The authors use the inflection points in order to identify the active and inactive modules to reduce the complexity of the nonlinear equation of each string, thus

reducing the calculation burden of the numerical method that solves the equation. Moreover, the inflection points are also used in order to restrict the search range for the solution of the string current, which considerably reduces the solution time.

The modeling procedures that were introduced in [14,15] allow for modeling a PV array with any size and connected in any connection scheme. Both procedures use the single diode model to represent the modules in a PV panel, and they are based on the circuital nodes and meshes principles to define a system of nonlinear equations that represents the array. That system of equations is solved to finally obtain the global output current of the array for a given voltage; then, performing a voltage sweep, it is possible to calculate the array I-V and P-V curves. However, the solutions of the models proposed in [14,15] require a high number of both mathematical operations and numerical method iterations due to the large search space (voltage and current range), in which the solution must be searched for; hence, the calculation times could be long. Therefore, that general solution could be impractical for some PV application such as: dynamic or static reconfiguration including heuristic methods [16,17] in which the calculation of the power provided by the array must be done in short time; validation and evaluation of Maximum Power Point Tracking (MPP) techniques, as the ones introduced in [18,19], in which optimization algorithms are implemented in order to improve the performance of the control stage in PV systems; or even to provide an optimal design for large PV plants. With the procedure that is presented in [14], a $10 \times 5$ irregular PV array takes 10 min. and 8 s to be solved for a given irradiance and temperature conditions, while by using the procedure introduced in [15], the time is reduced to 5 min. and 18 s. Despite the improvement in the execution time, in a planning of PV systems or reconfiguration scenario, it could be impractical if several configurations must be evaluated in order to define the best for a given operating condition; moreover, the execution times will increase if a larger array is evaluated.

In the previous literature analysis, two key points are identified. The first one is that there is a lack of fast modeling techniques able to analyze PV arrays in any configuration with any size, since the models proposed in [14,15] require a high number of mathematical operations that imply high execution times. Those high execution times make those models impractical for important applications like reconfiguration, MPPT, or PV array sizing. The second key point is that the inflection points concept has been used to reduce the solution time of SP arrays models; however, the inflection points concept has not been used for models of other PV array configurations or for the general models that are proposed in [14,15].

Therefore, this paper is based on the following hypothesis: it is possible to improve the computation times for analyzing PV arrays with any size and configuration by combining the inflection points modeling technique of [10] with the circuital nodes principle that is presented in [14]. In this way, the new solution provides the same analytical versatility to model any PV array (any size and configuration), but with much shorter processing times. This feature will be useful for reconfiguration techniques, analysis, and design of large PV fields, among others. Moreover, the adoption of the inflection points concept also improves the convergence rate of the general model, which reduced the prediction errors in comparison with the models that were reported in [14,15].

The rest of the paper is organized, as follows: Section 2 describes the mismatching conditions problem, and shows the use of different connection schemes in order to improve the power production; then, Section 3 describes the inflection point concept and usability. Section 4 presents the proposed method for calculating the inflection points of a PV array with any connection scheme, and Section 5 describes the use of those inflection points to efficiently calculate the PV current and power. The performance of the proposed model is evaluated in Section 6, where both the accuracy and speed of the proposed solution are contested with a reference model. Similarly, Section 7 illustrates the use of the proposed model in a reconfiguration application. Finally, the conclusions close the paper.

## 2. Mismatching Conditions and Internal Connections in PV Arrays

PV arrays are formed by multiple PV panels that are connected in both series and parallel schemes. Such a connection does not depends on the physical organization of the PV panels in the PV array; for example, the general PV array that is presented at the left of Figure 1 could be connected in different configurations. The same figure presents three electrical configurations that are commonly used in industrial applications: the series-parallel (SP), which considers the panels connected in independent strings, which are then connected in parallel; the total-cross-tied (TCT), which considers a complete interconnection of all the panels; the bridge-linked (BL), which considers a bridge-like structure.

Commercial PV panels include bypass diodes that are connected in anti-parallel with the modules, which protect the module from operating at the second quadrant, i.e., avoiding power consumption and artificial degradation under mismatching conditions, as is reported in [20]. An active bypass diode short-circuits the associated PV module; hence, different connections between the PV modules produce different voltages and currents into the modules, which affect the power that is produced by the PV array. Therefore, depending on the shading profile covering the PV array, some of those connections could provide higher power, as it is discussed in [6]. Therefore, it is convenient to select the connections that enable the PV array to produce the highest power possible. Such a challenge has been addressed in several works. For example, in [21], it was concluded that SP arrays produce higher power for shading profiles, in which all of the array modules experience different levels of irradiance, but the first column has lower levels; while, TCT arrays produce higher power for shading profiles, such short narrow, short wide, or diagonal with irradiance variations from 1000 W/m$^2$ to 400 W/m$^2$ [22]. However, according to [23] for combinations of different shading cases such short and long wide at irradiance levels from 200 W/m$^2$ to 900 W/m$^2$, Su-Do-Ku configuration exhibits the best performance when compared to SP and TCT.

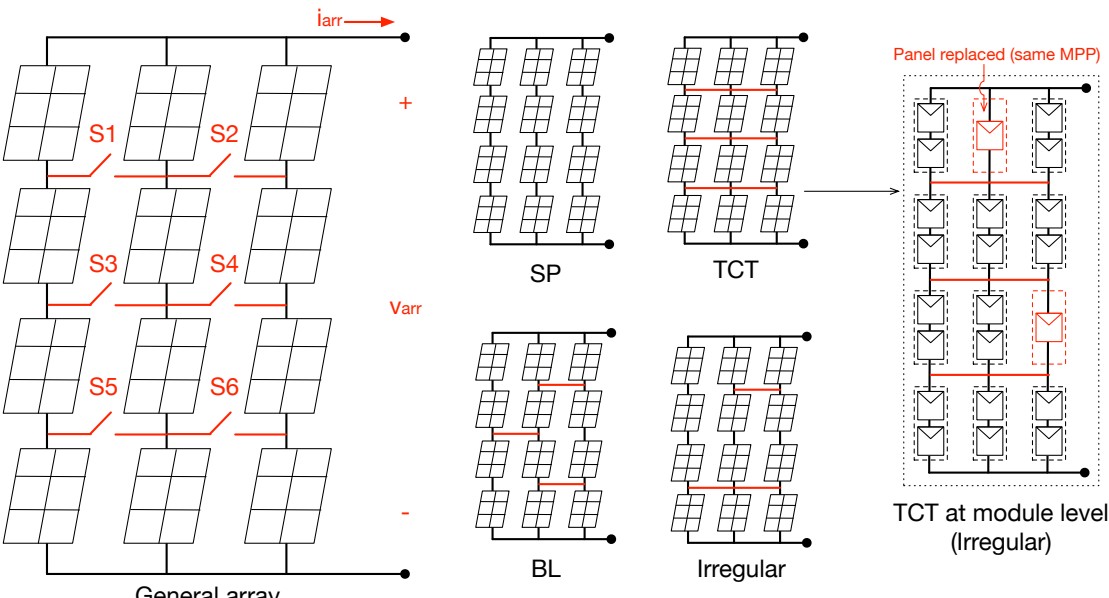

**Figure 1.** Examples of regular and irregular photovoltaic (PV) array configurations.

However, depending on the array location and the objects around the array, the shading profiles could have regular forms like rectangles or irregular forms with holes and branches. Such different shading conditions are presented in Figure 2 for a real PV installation, where the same PV array could have panels exposed to regular and irregular shades. Hence, under real operating conditions, it is difficult (or

even impossible) to define a regular connection (SP, TCT, BL, etc.), providing the highest power possible; instead, those classical configurations will produce a sub-optimal power production, which reduces the installation profitability.

Moreover, commercial PV installations are constructed while using commercial PV panels, which could be formed by one, two, or even three modules that are connected in series; hence, for example, an apparent TCT configuration formed by two-module panels (e.g., a BP585) will not be, in fact, a TCT structure. Besides, some manufacturers offer replacement panels for well-adopted references. For example, the ERDM 85SM/5 PV panel provides the same power and current as the BP585, but the ERDM 85SM/5 only has a single module. Therefore, it is possible to replace some BP585 (two-module panel) with ERDM 85SM/5 (single-module panel) without any inconvenience, but the resulting apparent TCT structure will be, in fact, an irregular configuration, as depicted in Figure 1. Therefore, modeling such real-application cases requires a general model for PV arrays, such as that proposed in [14,15].

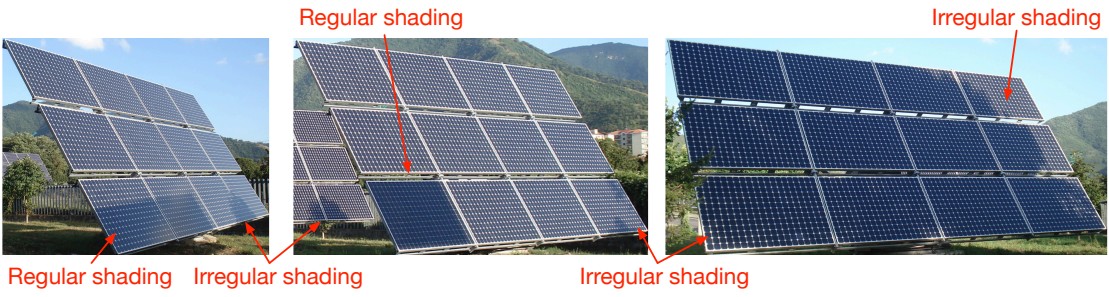

**Figure 2.** Regular and irregular shading patterns on PV arrays.

Furthermore, the work reported in [15] proposes adopting irregular connection schemes to the PV array to improve the power production under irregular shading conditions. Figure 1 shows an example of an irregular connection scheme (named Irregular), which does not follow any classical configuration (SP, TCT, BL, etc.). Such an irregular configuration enables the PV array of Figure 1 in order to provide a highest maximum power in comparison with the SP, TCT, and BL connection schemes for some irregular shading profiles, as depicted in Figure 3. This figure reports four irregular shading patterns, in which the shade intensity is reported as a percentage (0 % means no shading). The figure also reports the power production of the PV array with SP, TCT, BL, and the irregular connections. For the shading profile of Figure 3a, the irregular connections allow for the highest power production, followed by the SP option; while, the TCT and BL options produce the same power. Under the shading pattern 2 (Figure 3b) the irregular option also produces the highest power, while SP and TCT produce almost the same power, and BL produces the lowest power. For the shading pattern shown in Figure 3c, SP produces the lowest power, TCT and BL produce the same power, but again the irregular option provides the highest power production. Finally, for the shading pattern shown in Figure 3d, the irregular option also produces the highest power, followed by TCT, BL, and SP.

The previous simulations show that the same PV array, under shading conditions, will produce different power, depending on the connection scheme. Moreover, those results also confirm that configuring the array connections using an irregular scheme could increase the power production in comparison with the classical options (SP, TCT, BL, etc.). Therefore, the general PV array models that were proposed in [14,15] can be used to select the best configuration scheme (regular or irregular) improving the power production, which can be very useful for:

1.  Design the fixed connection of a PV array subjected to shading conditions.

2. Reconfigure the PV array in real-time to adjust the best connections, depending on the shading pattern, which changes, depending on the sun position.
3. Design a simulation platform to estimate the power production of any PV array with regular or irregular connections.

However, since there are multiple possibilities for irregular connections, and such a number increases with the size of the PV array, the first two applications require a fast PV model to provide practical calculation times. Similarly, for the third application, fast calculation times are also desirable. Therefore, the following sections describe a modeling approach that is designed to reduce the calculation time of the general PV model without reducing the model precision in comparison with [14,15].

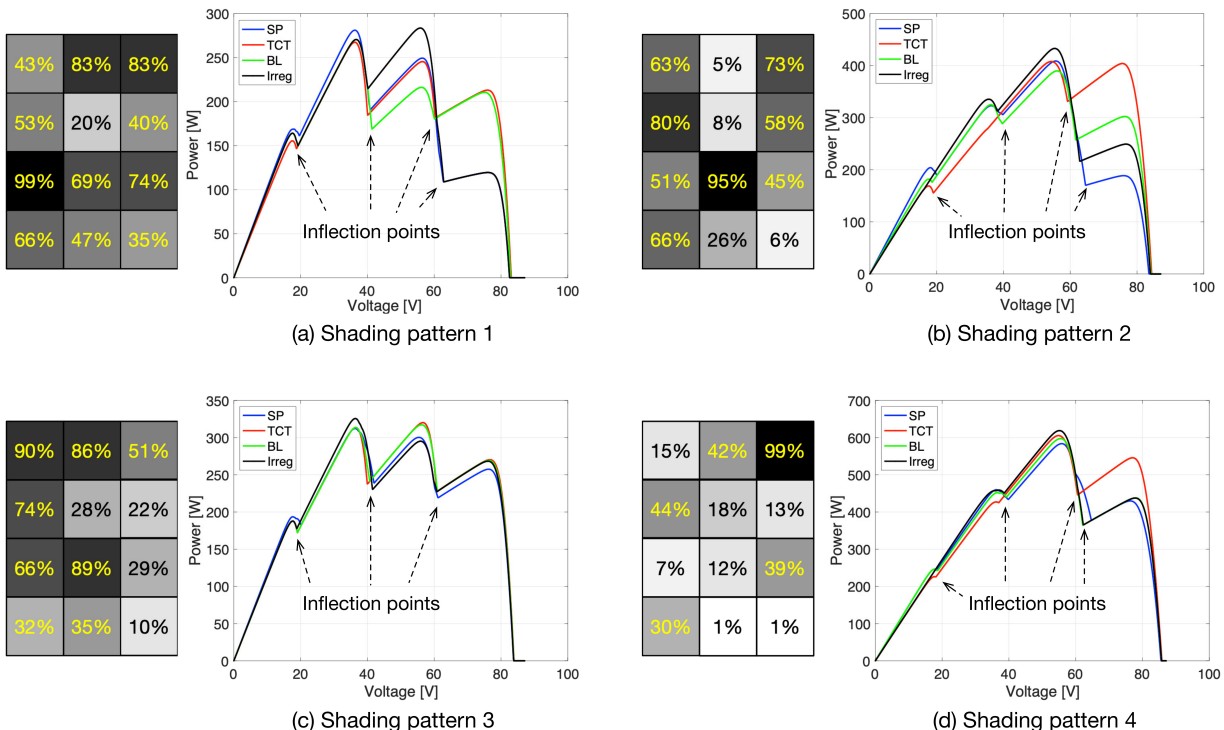

**Figure 3.** Power production of a PV array for different connections: SP, TCT, BL, and irregular.

## 3. Inflection Point Concept

The main concept that will be used for reducing the processing time of the general model is the inflection point in the I-V and P-V curves. Figure 3 shows the different inflection points occurring in the 12-module PV array operation due to different shading patterns, which activate some of the bypass diodes protecting the modules. The main objective of the PV array model is to calculate the current and power at a given array voltage, which is also needed in order to predict the I-V and P-V curves. Therefore, a previous calculation of the inflection points will allow to narrow the range of the possible solutions for the I-V and P-V data and, subsequently, reduce the processing time that is required by the model.

An inflection point in the I-V curve occurs when a bypass diode is activated. This concept is illustrated while using the two-module string depicted in Figure 4: the string is formed by modules 1 and 2, which are protected by the bypass diodes 1 and 2. In this example, module 2 is affected by a partial shade that reduces the its short-circuit current ($i_{sc2}$) to 40 % of the short-circuit current of module 1 ($i_{sc1}$), which is fully irradiated. The figure depicts, at the bottom-left, the I-V curves of both modules, where the effect of the shade on the second module is observed.

However, the operating points of both modules depend on the string voltage $v_{st}$ and current $i_{st}$. Moreover, the activations of the bypass diodes depend on the string and modules currents:

- If the string current is lower than the short-circuit current of both modules, i.e., $i_{st} < i_{sc2} < i_{sc1}$, which means that both modules could produce the string current, i.e., $i_{pv1} = i_{pv2} = i_{st}$, which imposes null current to both bypass diodes, i.e., $i_{d1} = i_{d2} = 0 \, [A]$. Therefore, both bypass diodes are inactive, and both modules are active to produce usable power. For the example of Figure 4, when $i_{st} < i_{sc2}$, both of the modules are active, despite having different short-circuit currents.

- Instead, for the same irradiance and shading pattern, if the string current is higher than the short-circuit current of some modules, i.e., $i_{sc2} < i_{st} < i_{sc1}$, this means that some modules are not able to produce the string current, i.e., $i_{pv2} = i_{sc2} < i_{st}$ and $i_{pv1} = i_{st}$, which forces the activation of the corresponding bypass diodes, i.e., $i_{d1} = i_{st} - i_{sc2} > 0$ and $i_{d2} = 0$. Therefore, some of the modules are inactive to produce usable power, i.e., in the example of Figure 4 the module 2 is short-circuit (inactive) when $i_{st} > i_{sc2}$.

- Finally, if the string current is higher than the short-circuit current of all the modules, i.e., $i_{st} > i_{sc1} > i_{sc2}$, this means that all of the modules are not able to produce the string current, which forces the activation of all the bypass diodes. Therefore, the PV array is inactive in producing usable power.

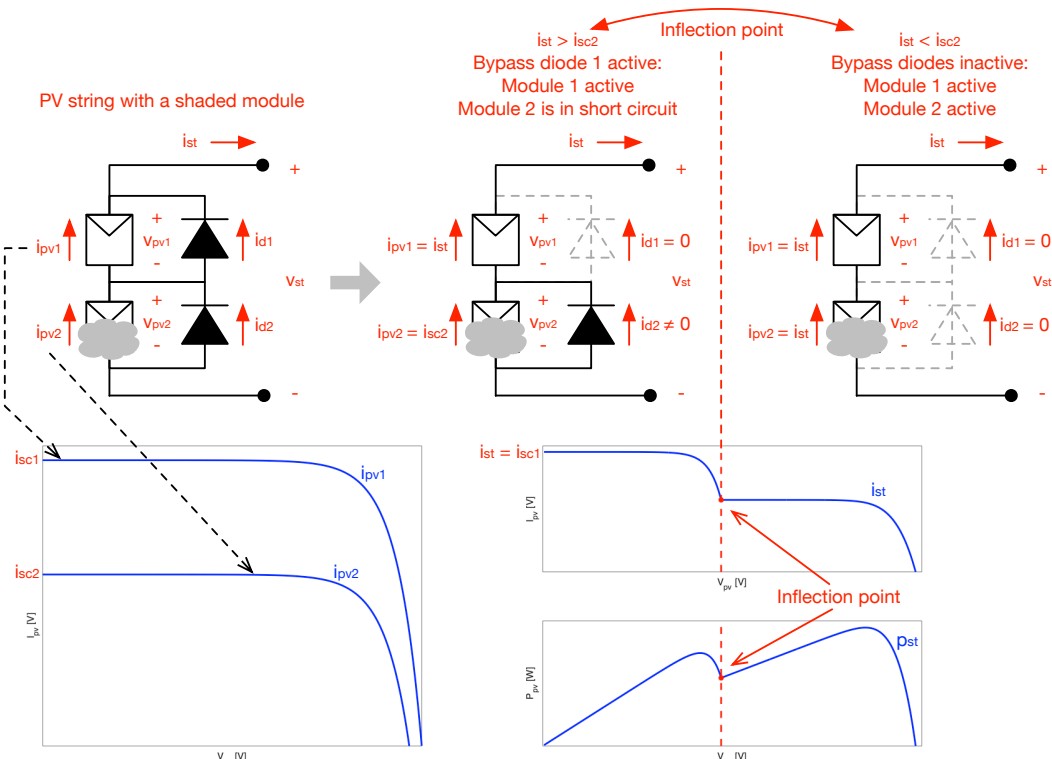

**Figure 4.** Inflection point concept.

In conclusion, the operating condition in which one (or multiple) bypass diode state changes (from active to inactive or vice versa) is known as an inflection point. The currents of the inflection point will enable limiting the possible values of the modules voltages for a given string current, which reduces the search space for the equations solver, as it will be shown in Sections 5–7. Finally, the following section

presents a modeling approach to calculate the inflection points for a general PV array with any connection scheme, i.e., regular or irregular connections.

## 4. Inflection Points Calculation

This section presents a procedure for calculating the inflection point voltages, also known as inflection voltages, in a PV array with any configuration; an example is also presented in order to illustrate the application of the proposed procedure. The section begins with the explanation of the module model and it continues with the matrix representation of the PV array and its sub-arrays. These two elements are required for explaining the proposed procedure and example.

### 4.1. PV Module Representation

In this paper, each PV module is represented by the single-diode model, since it provides a tradeoff between accuracy and complexity [3]. The equivalent circuit of the single-diode model, including the bypass diode, is shown in Figure 5, where the current source ($I_{ph}$) represents the current that is generated by the PV effect, the diode $D$ includes the nonlinear behavior of the PV cells and the resistors $R_h$ and $R_s$ represent the leakage currents and ohmic losses, respectively. Moreover, the bypass diode ($D_{bd}$) is connected at the output terminals of the PV module in order to provide an alternative path for the current when the module is shaded [20].

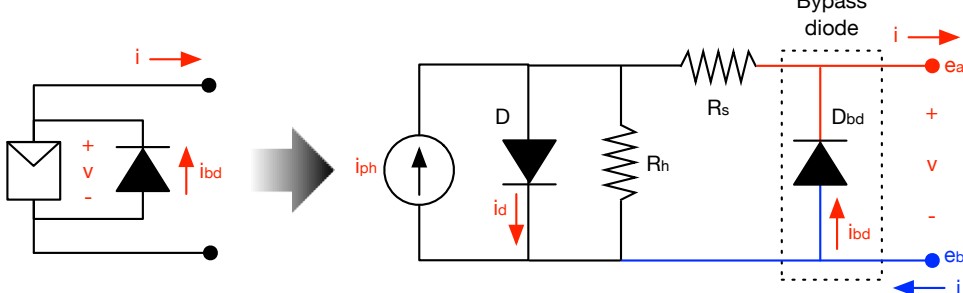

**Figure 5.** PV module single-diode model equivalent circuit, including the bypass diode.

The module output current ($I$) can be expressed as a function of the node voltages at the positive ($e_a$) and negative ($e_b$) terminals, as shown in (1), where $N_s$ is the number of series connected cells that form the module, $W_0(\cdot)$ is the LambertW function, $I_{sat}$ and $I_{sat,bd}$ are the inverse saturation current of diodes $D$ and $D_{bd}$, respectively, and $V_t$ and $V_{t,bd}$ are the termal voltage of diodes $D$ and $D_{bd}$, respectively. In turn, $V_t = n \cdot k \cdot T/q$ and $V_{t,bd} = n_{bd} \cdot k \cdot T/q$, where $n$ and $n_{bd}$ are the ideality factors of $D$ and $D_{bd}$, respectively, $k$ is the Boltzmann constant, $q$ is the electron charge, and $T$ is the module temperature in Kelvin [9].

$$I = -\frac{N_s \cdot V_t \cdot W_0(\theta)}{R_s} + I_{sat,bd} \cdot \left[\exp\left(\frac{-(e_a - e_b)}{V_{t,bd}}\right)\right] -$$

$$I_{sat,bd} + \frac{R_h \cdot \left(I_{ph} + I_{sat}\right) - (e_a - e_b)}{R_h + R_s}$$

$$\theta = \left(\frac{R_h \cdot R_s}{R_h + R_s}\right) \cdot \frac{I_{sat}}{N_s \cdot V_t} \cdot \exp\left[\frac{(R_s + R_h) \cdot \left(I_{ph} + I_{sat}\right) + R_h \cdot (e_a - e_b)}{N_s \cdot V_t \cdot (R_s + R_h)}\right]$$

(1)

On the one hand, the five parameters of the SDM (i.e $I_{ph}$, $I_{sat}$, $n$, $R_s$, and $R_h$) can be calculated from the datasheet information or from experimental I-V curves by using the procedures proposed in [24,25], respectively. On the other hand, the two parameters of the bypass diode (i.e., $I_{sat,bd}$, $n_{bd}$) can be calculated from the experimental measurements, as shown in [20], or from the information in the bypass diode datasheet. In general, the $I_{ph}$ depends on the module irradiance and temperature, $I_{sat}$ and $I_{sat,bd}$ vary with the module temperature, and $n$, $n_{bd}$, $R_s$, and $R_h$ can be assumed to be constant [3].

When the PV module is in short-circuit, the current that is provided by the module ($I_{sc}$) is calculated from (1) replacing $(e_a - e_b) = 0 \ [V]$. The expression for calculating $I_{sc}$ is introduced in (2), and it is an important parameter that is used in the following sections to calculate the inflection voltages.

$$I_{sc} = -\frac{N_s \cdot V_t \cdot W_0\left(\theta_{sc}\right)}{R_s} + \frac{R_h \cdot \left(I_{ph} + I_{sat}\right)}{R_h + R_s} \tag{2}$$

$$\theta_{sc} = \left(\frac{R_h \cdot R_s}{R_h + R_s}\right) \cdot \frac{I_{sat}}{N_s \cdot V_t} \cdot \exp\left[\frac{(R_s + R_h) \cdot \left(I_{ph} + I_{sat}\right)}{(R_s + R_h) \cdot N_s \cdot V_t}\right]$$

### 4.2. Matrix Representation a PV Array and Its Sub-Arrays

A PV array that formed by $M$ columns and $N$ rows, i.e., an $N \times M$ array, in any configuration can be represented by a set of matrices. One $N \times M$ matrix for each parameter of the SDM and the $D_{bd}$ ($M_{Iph}$, $M_{Isat}$, $M_n$, $M_{Rs}$, $M_{Rh}$, $M_{Isatbd}$, $M_{nbd}$), one $N \times M$ matrix for the short-circuit currents ($M_{Isc}$), and one $(N-1) \times (M-1)$ matrix (named $M_{conn}$) in order to represent the connections between the modules of the PV array.

The elements of $M_{conn}$ can be 1 or 0 in order to represent whether there is or not a tie between two consecutive strings of the array. Hence, a 1 in the row $r$ and column $c$ of $M_{conn}$ ($M_{conn}(r,c) = 1$) indicates that the negative terminal of the modules in row $r$ and column $c$ (module $(r,c)$) is connected to the negative terminal of the module in row $r$ and column $c+1$ (module $(r,c+1)$). Instead, $M_{conn}(r,c) = 0$ means that there is no connection between the negative terminals of modules $(r,c)$ and $(r,c+1)$.

According to [14], a sub-array is defined as a string or set of strings in the array that are connected to the rest of the array only in the ground and $V_{arr}$ nodes, i.e., the lower and upper nodes. Therefore, there are no additional connections to the left or right of the sub-array. From this definition, it is possible to identify the sub-arrays by finding the columns of zeros in $M_{conn}$; thus, each column of zeros in $M_{conn}$ indicates the end of a sub-array and the beginning of the next one. The number of sub-arrays in a $N \times M$ array is defined in (3), where $N_z$ is the number of columns of zeros in $M_{conn}$ [14].

$$N_{sa} = N_z + 1 \tag{3}$$

A vector $C_z$ is defined, as shown in (4), in order to identify the first and last column of each sub-array, where the first and last elements are 0 and $M$, respectively, and the other elements ($Z_{c,1} \cdots Z_{c,N_z}$) are the column number of each column of zeros in $M_{conn}$. Subsequently, it is possible to define the first ($SA_{fc,i}$) and last column ($SA_{lc,i}$) of the sub-array $i$, as illustrated in (5) [14].

$$C_z = \begin{bmatrix} 0 & Z_{c,1} & \ldots & Z_{c,i} & \ldots & Z_{c,N_{sa}} & M \end{bmatrix} \tag{4}$$

$$SA_{fc,i} = C_z(i) + 1, \quad SA_{lc,i} = C_z(i+1) \tag{5}$$

From $SA_{fc,i}$ and $SA_{fl,i}$, it is possible to define the matrices that contain the parameters of the sub-array $i$ ($M_{Iph,i}^{sa}$, $M_{Isat,i}^{sa}$, $M_{Vt,i}^{sa}$, $M_{Rs,i}^{sa}$, $M_{Rh,i}^{sa}$, $M_{Isatbd,i}^{sa}$, $M_{Vtbd,i}^{sa}$, and $M_{Isc,i}^{sa}$) from the matrices of the array parameters,

as introduced in (6), taking into account that all of the sub-arrays have $N$ rows. Finally, the connection matrix of sub-array $i$ ($M_{conn,i}^{sa}$) is defined in (7), which takes into account that it has one row less and one column less than the sub-array [14].

$$
\begin{aligned}
M_{Iph,i}^{sa} &= M_{Iph}(r,c),\ M_{Isat,i}^{sa} = M_{Isat}(r,c),\ M_{Vt,i}^{sa} = M_{Vt}(r,c),\\
M_{Rs,i}^{sa} &= M_{Rs}(r,c),\ M_{Rh,i}^{sa} = M_{Rh}(r,c),\ M_{Isatbd,i}^{sa} = M_{Isatbd}(r,c),\ M_{Isc,i}^{sa} = M_{Isc}(r,c)\\
M_{Vtbd,i}^{sa} &= M_{Vtbd},\ \forall\, r \in [1 \ldots N],\ c \in [SA_{fc,i} \ldots SA_{lc,i}] \\
M_{conn,i}^{sa} &= M_{conn}(r_c, c_c),
\end{aligned}
\tag{6}
$$

$$
\forall\, r_c \in [1 \ldots (N-1)],\ c_c \in [SA_{fc,i} \ldots (SA_{lc,i}-1)]
\tag{7}
$$

*4.3. Inflection Voltage Calculation for a Sub-Array*

An inflection point in the I-V curve of a sub-array is produced by the activation/deactivation of the bypass diode of a sub-array's module. In general, if the current flowing through the module in the row $i_o$ and column $j_o$ is higher than its short-circuit current ($I_{sc,i_o,j_o}$); then, the bypass diode is activated to provide an alternative path for the current in excess; otherwise, the bypass diode is inactive (reverse biased) and no current flows through it. The sub-array voltage when the module $(i_o, j_o)$ is in short-circuit (i.e., $I_{i_o,j_o} = I_{sc,i_o,j_o}$) is defined as the inflection voltage ($V_{o,i_o,j_o}$) that us produced by the module $(i_o, j_o)$. While taking into account that all of the currents in the sub-array decrease as the sub-array voltage increases, the bypass diode of module $(i_o, j_o)$ is active if the sub-array voltage is lower than $V_{o,i_o,j_o}$ (i.e., ($V_{sa} \leq V_{o,i_o,j_o}$)), and it is inactive otherwise.

In order to calculate $V_{o,i_o,j_o}$, it is necessary to solve the sub-array node voltages, which can be performed independently from the other sub-arrays, as explained in the previous section. The number of nodes in a sub-array ($N_n$), excluding the ground node (bottom node) and the array voltage ($V_{arr}$) node (top node), can be calculated while using (8), where the left term corresponds to the number of node in an SP configuration and the right term is the sum of all the elements in $M_{conn}^{sa}$. Hence, when considering the top node voltage as unknown (i.e., $V_{arr}$ is unknown), a sub-array has $N_n + 1$ unknown node voltages, i.e., $e_j \,\forall\, j \in [1 \cdots N_n + 1]$, which form the unknowns vector, named $\vec{V}_n$. It is important to note that the top node voltage is $e_{N_n+1}$, while the others are numbered from top to bottom and from left to right in the sub-array.

$$
N_n = (N-1) \cdot M_{sa} - \sum_{k_2=1}^{M_{sa}} \sum_{k_1=1}^{N} M_{conn}^{sa}(k_1, k_2)
\tag{8}
$$

The system of $N_n + 1$ nonlinear equations (named $F_o(\vec{V}_n)$) that are required to calculate $\vec{V}_n$ is obtained by applying the KCL in the first $N_n$ nodes and from the $0\,V$ restriction imposed between the positive and negative terminals of the module in short-circuit (i.e., $I_{i_o,j_o} = I_{sc,i_o,j_o}$ then $e_a(i_o,j_o) - e_b(i_o,j_o) = 0\,V$). In order to apply KCL in the first $N_n$ nodes of the sub-array, it is necessary to identify the node voltages at the positive and negative terminal of each module in the sub-array. This identification is expressed in an $N \times M_{sa}$ matrix ($M_{nvo}^{sa}$), where the element at row $r$ and column $c$ of $M_{nvo}^{sa}$ (e.g., $M_{nvo}^{sa}(r,c) = k$) indicates that the voltage at the positive terminal of the module $(r,c)$ is $V_n(k)$, or $e_a(r,c) = \vec{V}_n(k) = e_k$. Moreover, $M_{nvo}^{sa}(r,c) = k$ also indicates the node voltage at the negative terminal of the module $(r-1,c)$, since the negative terminal of the module $(r-1,c)$ is connected to the positive terminal of the module $(r,c)$; hence, $M_{nvo}^{sa}(r,c) = k$ also indicates that $e_b(r-1,c) = \vec{V}_n(k) = e_k$ if $r > 1$.

Once the voltages at positive and negative terminals of all the modules have been defined with $M_{nvo}^{sa}$, it is possible to apply KCL to the first $N_n$ nodes of the sub-array, where the current of each module is calculated with (1). Moreover, the last equation is obtained from the module in short-circuit condition

(module $(i_o, j_o)$), where $I_{i_o,j_o} = I_{sc.i_o,j_o}$ and $e_a(i_o, j_o) - e_b(i_o, j_o) = 0 \, [V]$. Such a condition implies that the voltages of the nodes at the positive and negative terminals of the module $(i_o, j_o)$ are the same, as shown in (9).

$$e_a(i_o, j_o) - e_b(i_o, j_o) = 0 \text{ or } \vec{V}_n(k_a) - \vec{V}_n(k_b) = 0 \tag{9}$$
$$\text{where } M^{sa}_{nvo}(i_o, j_o) = k_a \text{ and } M^{sa}_{nvo}(i_o + 1, j_o) = k_b$$

The connection pattern of each sub-array is defined by $M^{sa}_{conn}$, as explained in the previous section. Therefore, the structure of $F_o(\vec{V}_n)$ is not defined by a fixed structure; instead, it is defined by an algorithm that describes how to evaluate $F_o(\vec{V}_n)$ for a given $\vec{V}_n$, which is introduced as pseudocode in Algorithm 1. In such an algorithm, $i_o$ and $j_o$ are the row and column of the module in short-circuit (i.e., $I_{i_o,j_o} = I_{sc,i_o,j_o}$) and $i_f, l_m, r_m, n_n, n_1, n_2$ are auxiliary variables used to simplify the pseudocode description. The two iterative (*for*) loops in lines 2 and 3 go over all the sub-array nodes to apply KCL in each one of them. When a new node is found (line 4), the columns of the leftmost ($l_m$) and rightmost ($r_m$) modules that are connected to the node are identified. Subsequently, the *for* loop defined in line 6 evaluates the KCL for all of the modules connected to that node, column-by-column. Lines 7 to 13 and 15 to 25 calculate the current of the module that is connected above and below the analyzed node, respectively, while lines 14 and 26 adds and subtracts the modules' currents in order to construct the KCL colum- by-column. Lines 1 to 31 allow for the evaluation of the first $N_n$ elements of $F_o(\vec{V}_n)$; while, lines 32 to 38 evaluate the last element of $F_o(\vec{V}_n)$, which shows that the node voltages at the positive and negative terminals of the module in short-circuit condition (i.e., module $(i_o, j_o)$) are the same.

The evaluation of $F_o(\vec{V}_n)$, with Algorithm 1, is used by a numerical method in order to solve $\vec{V}_n$. From such a solution, the inflection voltage that is produced by the module in row $i_o$ and column $j_o$ (i.e., $V_{o,i_o,j_o}$) is defined in (10). Moreover, the nodes voltages $\vec{V}_n(k) \; \forall \, k \in [1 \cdots N_n]$ are defined as node inflection voltages that are produced by the module $(i_o, j_o)$, as shown in (11).

$$V_{o,i_o,j_o} = \vec{V}_n(N_n + 1) \tag{10}$$
$$V^k_{no,i_o,j_o} = \vec{V}_n(k) \; \forall \, k \in [1 \cdots N_n] \tag{11}$$

---

**Algorithm 1** Calculation of $F_o(\vec{V}_n)$

---

**INPUT:** $\vec{V}_n$, $N_n$, $M_{nvo}^{sa}$, $M_{conn}^{sa}$, $M_{Isc}^{sa}$, $i_o$, $j_o$, $N$, $M_{sa}$, sub-array parameters' matrices
**OUTPUT:** $F_o(\vec{V}_n)$

1: Set $i_f = 1$
2: **for** columns $j = 1$ **to** $M_{sa}$ **do**
3:    **for** rows $i = 1$ **to** $N - 1$ **do**
4:       **if** $j = 1$ **OR** $M_{conn}^{sa}(i, j - 1) = 0$ **then**:
5:          Identify $l_m$ and $r_m$ from $M_{conn}^{sa}$
6:          **for** modules in the node $j_c = l_m$ **to** $r_m$ **do**
7:             Set $n_n = M_{nvo}^{sa}(i, j_c)$ and $e_a(i, j_c) = \vec{V}_n(n_n)$
8:             Set $n_n = M_{nvo}^{sa}(i + 1, j_c)$ and $e_b(i, j_c) = \vec{V}_n(n_n)$
9:             **if** $i = i_o$ **AND** $j = j_o$ **then**:
10:                Set $I_{i,j_c} = M_{Isc}^{sa}(i, j_c)$
11:             **else**:
12:                Calculate $I_{i,j_c}$ with (1)
13:             **end if**
14:             Calculate $F_o(i_f) = F_o(i_f) + I_{i,j_c}$
15:             Set $n_n = M_{nvo}^{sa}(i + 1, j_c)$, $e_a(i + 1, j_c) = \vec{V}_n(n_n)$
16:             **if** $i = N - 1$ **then**:
17:                $e_b(i + 1, j_c) = 0$ [V]
18:             **else**:
19:                Set $n_n = M_{nvo}^{sa}(i + 2, j_c)$ and $e_b(i + 1, j_c) = \vec{V}_n(n_n)$
20:             **end if**
21:             **if** $i + 1 = i_o$ **AND** $j = j_o$ **then**:
22:                Set $I_{i+1,j_c} = M_{Isc}^{sa}(i, j_c)$
23:             **else**:
24:                Calculate $I_{i+1,j_c}$ with (1)
25:             **end if**
26:             Calculate $F_o(i_f) = F_o(i_f) - I_{i+1,j_c}$
27:          **end for** modules in the node
28:          Set $i_f = i_f + 1$
29:       **end if**
30:    **end for** rows
31: **end for** columns
32: **if** $i_o < N$ **then**:
33:    Set $n_{n1} = M_{nvo}^{sa}(i_o, j_o)$ and $n_{n2} = M_{nvo}^{sa}(i_o + 1, j_o)$
34:    Calculate $F_o(N_n + 1) = \vec{V}_n(n_{n1}) - \vec{V}_n(n_{n2})$
35: **else**:
36:    Set $n_n = M_{nvo}^{sa}(i_o, j_o)$
37:    Calculate $F_o(N_n + 1) = \vec{V}_n(n_n)$
38: **end if**
39: **Return** $F_o(\vec{V}_n)$

---

When considering that all of the modules in the sub-array have bypass diodes, and that the sub-array has $N$ rows and $M_{sa}$ columns, it is necessary to define an $N \times M_{sa}$ matrix with all the sub-array inflection voltages ($M_{Vo}$), where $M_{Vo}(i, j) = V_{o,i,j}$, as shown in (12). Similarly, the node inflection voltages are also defined in $N_n$ matrices (from $M_{Vno}^1$ to $M_{Vno}^{N_n}$), as illustrated in (13).

$$
M_{Vo} = \begin{bmatrix} V_{o,1,1} & \cdots & V_{o,1,M_{sa}} \\ \vdots & \ddots & \vdots \\ V_{o,N,1} & \cdots & V_{o,N,M_{sa}} \end{bmatrix} \tag{12}
$$

$$
M_{Vno}^{k} = \begin{bmatrix} V_{no,1,1}^{k} & \cdots & V_{no,1,M_{sa}}^{k} \\ \vdots & \ddots & \vdots \\ V_{no,N,1}^{k} & \cdots & V_{no,N,M_{sa}}^{k} \end{bmatrix} \quad \forall\, k \in [1 \cdots N_n] \tag{13}
$$

$M_{Vo}$ is transformed into a vector and sorted ascendantly ($\vec{V}_{Vo}$) to locate the inflection voltages along the sub-array I-V curve. The matrices $M_{Vno}^{k} \ \forall\, k \in [1 \cdots N_n]$ are also transformed into vectors and sorted following the same order of $\vec{V}_{Vo}$ in order to obtain $N_n$ vectors: $\vec{V}_{Vno}^{k} \ \forall\, k \in [1 \cdots N_n]$. Subsequently, $\vec{V}_{Vo}$ and the node inflection voltage vectors can be used in order to define the solution search range of the $N_n$ node voltages for a given sub-array voltage, as will be explained in the following section. Finally, the flow chart that is shown in Figure 6 summarizes the procedure to calculate the sub-array and node inflection voltages.

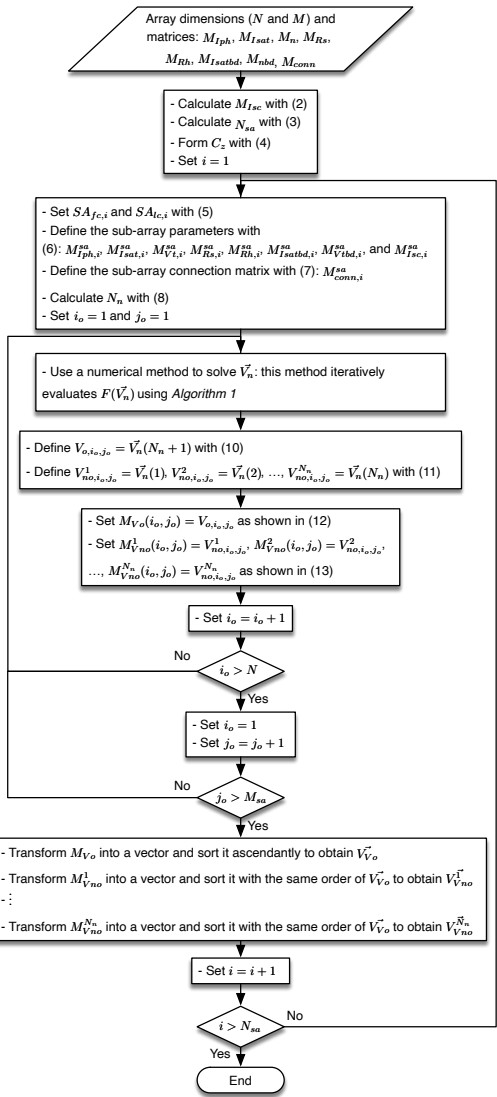

**Figure 6.** Flow chart of the proposed procedure to calculate the inflection voltages of a sub-array.

### 4.4. Calculation Example

The procedure described in this section is applied to the $3 \times 3$ array shown in that is Figure 7 to illustrate its application. For the example, the PV panel considered is an ERDM 85SM/5, which is formed by one module of 36 cells ($N_s = 36$). The standard test condition (STC) parameters of such a panel are: $I_{sc,STC} = 5.13$ [A], $I_{mpp,STC} = 4.80$ [A], $V_{oc,STC} = 21.78$ [V], $V_{mpp,STC} = 17.95$ [A], $\alpha_{Isc} = 0.02\%$, and $\alpha_{Voc} = -0.37\%$, where the subscripts $mpp$ and $oc$ indicate maximum power point and open-circuit, respectively; while, $\alpha_{Isc}$ and $\alpha_{Voc}$ are the short-circuit current and open-circuit voltage temperature coefficients. The following SDM parameters are obtained while using the STC parameters and the procedure proposed in [24]: $I_{ph} = 5.13$ [A], $I_{sat} = 1.1841$ [ηA], $V_t = 981.5$ [mA], $R_s = 186.4$ [mΩ], and $R_h = 261.099$ [mΩ]. Moreover, the bypass diode parameters were selected as $I_{sat,bd} = 1.00$ [μA], $V_{t,bd} = 6.9$ [mV].

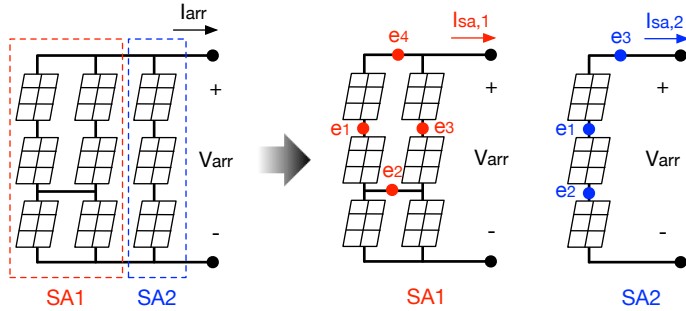

**Figure 7.** Example of a $3 \times 3$ array with 2 sub-arrays.

This array has three rows ($N = 3$) and three columns ($M = 3$) and, for the sake of simplicity, it is assumed that all of the modules have the same parameters, the cells temperature is 25 °C, and the global irradiance is $1 \, \text{kW/m}^2$. Moreover, the array is described by eight $3 \times 3$ matrices ($M_{Iph}$, $M_{Isat}$, $M_n$, $M_{Rs}$, $M_{Rh}$, $M_{Isatbd}$, $M_{nbd}$, and $M_{Isc}$), and the connection matrix ($M_{conn}$), which has 2 rows and 2 columns. $M_{Isc}$ is shown in (14) as an example of the eight $3 \times 3$ matrices with the array parameters; while, (15) shows $M_{conn}$ for the analyzed array. It can be seen that $M_{conn}(2,1) = 1$, since there is a connection between the negative terminals of modules in the second row between columns 1 and 2 (see Figure 7). The other elements of $M_{conn}$ are 0, because there are no additional connections among the array columns.

$$M_{Isc} = \underbrace{\begin{bmatrix} 4.620 & 4.106 & 2.566 \\ 3.593 & 3.080 & 2.566 \\ 2.566 & 2.053 & 5.133 \end{bmatrix}}_{M_{Isc,1}^{sa} \quad M_{Isc,2}^{sa}} [\text{A}] \tag{14}$$

$$M_{conn} = \underbrace{\begin{bmatrix} 0 & 0 \\ 1 & 0 \end{bmatrix}}_{M_{conn,1}^{sa} \; M_{conn,2}^{sa}} \tag{15}$$

In (15), the second column only has zeros (i.e., $N_z = 1$ and $Z_{c,1} = 2$); then, there are two sub-arrays ($N_{sa} = 2$) and the resulting vector $C_z$ is shown in (16). From $C_z$, are obtained the first and last columns of sub-array 1 ($SA1$) and sub-array 2 ($SA2$), as given in (17) and (18), respectively. Moreover, with (17) and (18), it is possible to define the matrices of each sub-array by using (6) and (7). As an example, $M_{Isc,1}^{sa}$, $M_{Isc,2}^{sa}$, $M_{conn,1}^{sa}$, and $M_{conn,2}^{sa}$ are illustrated in (14) and (15) while using the horizontal braces at the bottom of the matrices.

$$C_z = \begin{bmatrix} 0 & 2 & 3 \end{bmatrix} \tag{16}$$

$$SA_{fc,1} = 1, \; SA_{lc,1} = 2 \tag{17}$$

$$SA_{fc,2} = 3, \; SA_{lc,2} = 3 \tag{18}$$

From Figure 7, it is observed that $SA1$ and $SA2$ have three and two nodes, respectively, excluding the ground and $V_{arr}$ nodes; hence, $N_n = 3$ for $SA1$ and $N_n = 2$ for $SA2$. For the inflection voltage calculation, the voltage at the top node is also unknown; therefore, $\vec{V}_n = [e_1 \, e_2 \, e_3 \, e_4]$ for $SA1$ and $\vec{V}_n = [e_1 \, e_2 \, e_3]$ for

*SA*2. While using $\vec{V}_n$ and $M_{conn}^{sa}$, it is possible to define the matrix $M_{nvo}^{sa}$ of each sub-array, as shown in (19), where $M_{nvo,1}^{sa}$ corresponds to *SA*1 and $M_{nvo,2}^{sa}$ corresponds to *SA*2.

$$M_{nvo,1}^{sa} = \begin{bmatrix} 4 & 4 \\ 1 & 3 \\ 2 & 2 \end{bmatrix}, \ M_{nvo,2}^{sa} = \begin{bmatrix} 3 \\ 1 \\ 2 \end{bmatrix} \tag{19}$$

At this point, all of the parameters of both sub-arrays are defined and the inflection voltages can be calculated. The sub-array and node inflection voltages that are generated by the module in the position $(1,1)$ of *SA*1 are calculated by using the Trust-Region Reflective method, which uses Algorithm 1 with $i_o = 1$ and $j_o = 1$ in order to find $\vec{V}_n = [-0.1964 \ -0.1006 \ -0.1964 \ -0.1964] \ [V]$; then, $V_{o,1,1} = -0.1964 \ [V]$, $V_{no,1,1}^1 = -0.1964 \ [V]$, $V_{no,1,1}^2 = -0.1006 \ [V]$, and $V_{no,1,1}^3 = -0.1964 \ [V]$. This process is repeated for all of the modules in *SA*1 to obtain $M_{Vo}$, $M_{Vno}^1$, $M_{Vno}^2$, and $M_{Vno}^3$, as shown in (20) and (21).

$$M_{Vo} = \begin{bmatrix} -0.1964 & -0.1964 \\ 19.3587 & 19.4545 \\ 40.1054 & 40.1057 \end{bmatrix} \tag{20}$$

$$M_{Vno}^1 = \begin{bmatrix} -0.1964 & -0.1964 \\ -0.0959 & -0.0037 \\ 19.7068 & 19.7070 \end{bmatrix} M_{Vno}^2 = \begin{bmatrix} -0.1006 & -0.1006 \\ -0.0959 & -0.0958 \\ -0.0000 & -0.0000 \end{bmatrix} M_{Vno}^3 = \begin{bmatrix} -0.1964 & -0.1964 \\ -0.1655 & -0.0958 \\ 19.6799 & 19.6801 \end{bmatrix} \tag{21}$$

Figure 8 provides a graphical representation of the calculation of $M_{Vo}$, $M_{Vno}^1$, $M_{Vno}^2$, and $M_{Vno}^3$ for *SA*1, which shows the iterative process followed in order to fill those matrices by considering a different module in short-circuit condition for each iteration. Such a procedure is performed for all of the sub-arrays forming the PV array.

Now, $M_{Vo}$ is sorted ascendantly to obtain $\vec{V_{Vo}} = [-0.1964 \ - 0.1964 \ 19.3587 \ 19.4545 \ 40.1054 \ 40.1057] \ [V]$, and following the same order, the vectors for all of the modules are obtained: $\vec{V_{Vno}^1} = [-0.1964 \ - 0.1964 \ - 0.0959 \ - 0.0037 \ 19.7068 \ 19.7070] \ [V]$, $\vec{V_{Vno}^2} = [-0.1006 \ - 0.1006 \ - 0.0959 \ - 0.0958 \ 0.0000 \ 0.0000] \ [V]$, and $\vec{V_{Vno}^3} = [-0.1964 \ - 0.1964 \ - 0.1655 \ - 0.0958 \ 19.6799 \ 19.6801] \ [V]$. It is worth noting that those vectors correspond to *SA*1. The corresponding vectors for *SA*2 are obtained by following the same procedure, as follows: $\vec{V_{Vo}} = [-0.2043 \ 20.5391 \ 20.5393] \ [V]$, $\vec{V_{Vno}^1} = [-0.1022 \ 20.5906 \ 20.5391] \ [V]$, and $\vec{V_{Vno}^2} = [0.0000 \ 20.5906 \ 20.5906] \ [V]$. Finally, those vectors are used in order to calculate the node voltages; such a procedure will be illustrated in the following section while using a numerical example based on the previous numerical vectors.

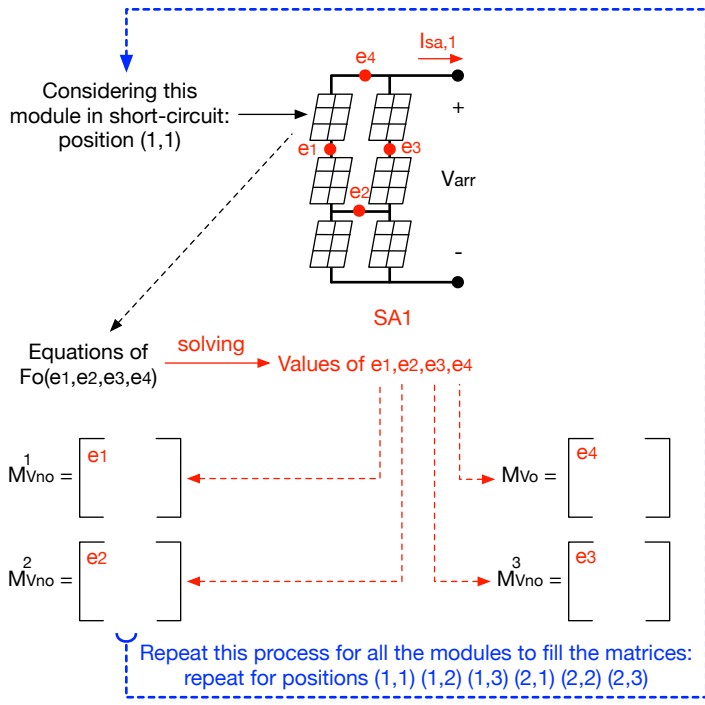

**Figure 8.** Graphical example of the calculation of $M_{Vo}$, $M^1_{Vno}$, $M^2_{Vno}$, and $M^3_{Vno}$ for $SA1$.

## 5. Sub-Array and Array Current Calculation Using Inflection Voltages

The main objective of the proposed model is to calculate the array current ($I_{arr}$) for a given array voltage ($V_{arr}$), which is imposed by the power converter where the array is connected to. When considering that each sub-array can be analyzed independently, this section explains the calculation process for a sub-array current, and the process to calculate $I_{arr}$ from the currents of all the sub-arrays.

### 5.1. Sub-Array Current Calculation

The objective of this section is to provide a procedure for calculating the sub-array node voltages, for a given $V_{arr}$, with a restricted search range in order to reduce the calculation time.

The number of nodes in a sub-array, excluding ground and $V_{arr}$ nodes, is defined in (8), naming the nodes from top to bottom and from left to right. Subsequently, it is necessary to construct a system of $N_n$ nonlinear equations ($F(\vec{V})$) by applying KCL in the $N_n$ nodes of the sub-array, where $\vec{V}$ is the vector formed by the $N_n$ unknown node voltages, i.e., $\vec{V} = [e_1 \cdots e_{N_n}]$. As explained in the previous section, the evaluation of $F(\vec{V})$ must be defined by an algorithm, since the structure of $F(\vec{V})$ is defined by $M^{sa}_{conn}$. Such an algorithm was proposed in [14] and it is described in Algorithm 2, where $i_f$, $l_m$, $r_m$, and $n_n$ are auxiliar variables that are used to simplify the pseudocode description. Moreover, the matrix $M^{sa}_{nv}$ contains the node voltage number of the negative terminals of the first $N-1$ modules in each column; therefore, $M^{sa}_{nv}(r,c) = k$ indicates that the node voltage at the negative terminal of the module in row $r$ and column $c$ is $\vec{V}(k)$, i.e., $e_b(r,c) = \vec{V}(k)$.

Algorithm 2 is used by a numerical method in order to evaluate $F(\vec{V})$ and calculate $\vec{V}$. The time that is required to obtain such a solution can be considerably reduced by providing a search range, for each element of $\vec{V}$, to the numerical method. Therefore, this paper proposes a method to define the search range of $\vec{V}$ from the sub-array and node inflection voltages.

---

**Algorithm 2** Calculation of $F(\vec{V})$

---

**INPUT:** $V_{arr}$, $\vec{V}$, $N_n$, $M_{nv}^{sa}$, $M_{conn}^{sa}$, $N$, $M_{sa}$, sub-array parameters' matrices
**OUTPUT:** $F(\vec{V}_n)$

1: Set $i_f = 1$
2: **for** columns $j = 1$ **to** $M_{sa}$ **do**
3:     **for** rows $i = 1$ **to** $N - 1$ **do**
4:       **if** $j = 1$ **OR** $M_{conn}^{sa}(i, j - 1) = 0$ **then**: Identify $lm$ and $rm$ from $M_{conn}^{sa}$
5:         **for** column $j_c = lm$ **to** $rm$ **do**
6:           Set $n_n = M_{nv}^{sa}(i, j_c)$, $e_b(i, j_c) = \vec{V}(n_n)$
7:           **if** $i = 1$ **then**: $e_a(i, j_c) = V_{arr}$
8:           **else**: Set $n_n = M_{nv}^{sa}(i - 1, j_c)$, and $e_a(i, j_c) = \vec{V}(n_n)$
9:           **end if**
10:          Calculate $I_{i,j_c}$ with (1) and $F_i(f_i) = F_i(f_i) + I_{i,j_c}$
11:          Set $n_n = M_{nv}^{sa}(i, j_c)$, $e_a(i + 1, j_c) = \vec{V}(n_n)$
12:          **if** $i = N - 1$ **then**: $e_b(i + 1, j_c) = 0$ [V]
13:          **else**: Set $n_n = M_{nv}^{sa}(i + 1, j_c)$, $e_b(i + 1, j_c) = \vec{V}(n_n)$.
14:          **end if**
15:          Calculate $I_{i+1,j_c}$ with (1) and $F_i(f_i) = F_i(f_i) - I_{i+1,j_c}$
16:         **end for**
17:         Set $i_f = i_f + 1$
18:       **end if**
19:     **end for** rows
20: **end for** columns
21: **Return** $F(\vec{V}_n)$

---

In a sub-array, the voltages of all the modules increase as $V_{arr}$ increases, since $V_{arr}$ is distributed along the modules in the sub-array columns. As consequence, all the sub-array node voltages also increase as $V_{arr}$ increases. Combining this behavior with the sub-array ($\vec{V}_{Vo}$) and node ($\vec{V}_{Vno}^k \;\forall\, k \in [1 \cdots N_n]$) inflection voltages, defined in the previous section, it is possible to restrict the search range of the sub-array node voltages for a given $V_{arr}$.

The first step is to identify the smallest sub-array inflection voltage ($\vec{V}_{Vo}(k)$) that is higher than $V_{arr}$, i.e., $V_{arr} < \vec{V}_{Vo}(k)$. Subsequently, when considering that $\vec{V}_{Vo}(k)$ is sorted ascendantly, it is possible to bound $V_{arr}$ with two sub-array inflection voltages: $\vec{V}_{Vo}(k-1) < V_{arr} < \vec{V}_{Vo}(k)$. Taking into account that the vectors of the node inflection voltages ($\vec{V}_{Vno}^k$) are organized in the same order of $\vec{V}_{Vo}$, then the search range of the node voltages is defined, as given in (22), where $e_1 \cdots e_{N_n}$ are the sub-array node voltages (excluding both top and bottom nodes), named with the same order used in the previous section, i.e., from top to bottom and from left to right.

$$
\begin{aligned}
V_{Vno}^{\vec{1}}(k-1) < e_1 < V_{Vno}^{\vec{1}}(k) \\
\vdots \\
V_{Vno}^{\vec{N}_n}(k-1) < e_{N_n} < V_{Vno}^{\vec{N}_n}(k)
\end{aligned}
\tag{22}
$$

The upper and lower bounds that are provided in (22) can be used by a numerical method to reduce the time for solving $\vec{V}$. Once $\vec{V}$ is solved, the sub-array current is calculated by adding the currents of the modules in the first row of the sub-array, as shown in (23).

$$I_{sa} = \sum_{c=1}^{M_{sa}} I(e_a(1,c), e_b(1,c)) \tag{23}$$

### 5.2. Calculation of the Array Current

The procedure that was described in the previous section is used to obtain the currents of all the sub-arrays. Subsequently, the array current is calculated as the sum of all the sub-arrays currents, as illustrated in (24), where $N_{sa}$ is the number of sub-arrays. Evaluating (24) provides a point in the array I-V curve for a given $V_{arr}$: $(I_{arr}, V_{arr})$; then, the array I-V curve can be constructed by performing a voltage sweep of $V_{arr}$ and calculating the corresponding values of $I_{arr}$. Finally, the P-V curve is easily obtained from the I-V data, which enables predicting the power production of the array.

$$I_{arr} = \sum_{k=1}^{N_{sa}} I_{sa,k} \tag{24}$$

### 5.3. Calculation Example

This example considers the same array that was introduced in Section 4.4, and the objective is to calculate $I_{arr}$ for $V_{arr} = 15$ [V] to illustrate the definition of the search bounds for the node voltages in each sub-array. SUbsequently, the example shows the I-V curves of $SA1$, $SA2$, and whole array, as well as the array inflection voltages for each sub-array.

In this example, $I_{arr} = I_{sa,1} + I_{sa,2}$ according to (24), therefore, the first step is to find $I_{sa,1}$ and $I_{sa,2}$ for $V_{arr} = 15$ [V]. In the $SA1$, the smallest sub-array inflection voltage that is higher than $V_{arr} = 15$ [V] is $\vec{V}_{Vo}(3) = 19.358$ [V]; then, $\vec{V}_{Vo}(2) < V_{arr} < \vec{V}_{Vo}(3)$ and $k = 3$. From this analysis, the search range of the node voltages is shown in (25), where the components of the vectors $V_{Vno}^{\vec{1}}$, $V_{Vno}^{\vec{2}}$ and $V_{Vno}^{\vec{3}}$ were defined in Section 4.4.

$$\begin{aligned} V_{Vno}^{\vec{1}}(2) &< e_1 < V_{Vno}^{\vec{1}}(3) \\ V_{Vno}^{\vec{2}}(2) &< e_2 < V_{Vno}^{\vec{2}}(3) \\ V_{Vno}^{\vec{3}}(2) &< e_3 < V_{Vno}^{\vec{3}}(3) \end{aligned} \tag{25}$$

In this paper, the vector of the node voltages $\vec{V}$ is calculated using the Sequential Quadratic Programming algorithm, which uses the search range that is defined in (25) and the Algorithm 2 to evaluate $F(\vec{V})$. For $V_{arr} = 15$ [V] the node voltages are $\vec{V} = [-0.1956 - 0.1004 - 0.1957]$ [V]. Subsequently, from $\vec{V}$ are calculated the node voltages at the positive and negative terminals of the modules in the first row, as follows: $e_a(1,1) = 15$ [V], $e_a(1,2) = 15$ [V], $e_b(1,1) = -0.1956$ [V], and $e_b(1,2) = -0.1957$ [V]. Subsequently, $I_{sa}$ is calculated using (23) to obtain $I_{sa,1} = 8.5764$ [A].

Following a similar procedure for $SA2$, it is obtained that $\vec{V} = [15.1019 \ 15.2039]$ [V], the node voltages for the module in the first row are $e_a(1,1) = 15$ [V] and $e_b(1,1) = 15.1019$ [V], which leads to $I_{sa,2} = 5.0553$ [A]. Finally, $I_{arr} = I_{sa,1} + I_{sa,2} = 13.6317$ [A].

The array I-V curve can be generated by repeating this process for different values of $V_{arr}$. The I-V curves of $SA1$, $SA2$, and the entire array, are depicted in Figure 9a, Figure 9b, and Figure 9c, respectively. In the I-V curves of both sub-arrays, the sub-array inflection voltages are marked with red circles while using the legend SAIVs.

From Figure 9, it is evident that constraining the search space to the voltage range between the closest inflection points, instead of the voltage range between 0 [V] and the open-circuit voltage of the array, provides much less options to the Sequential Quadratic Programming method, which improves the calculation time, since every option evaluated by that method will be much closer to the final solution. The following section provides a performance evaluation of this method in order to demonstrate the speed improvement that is provided by the proposed solution.

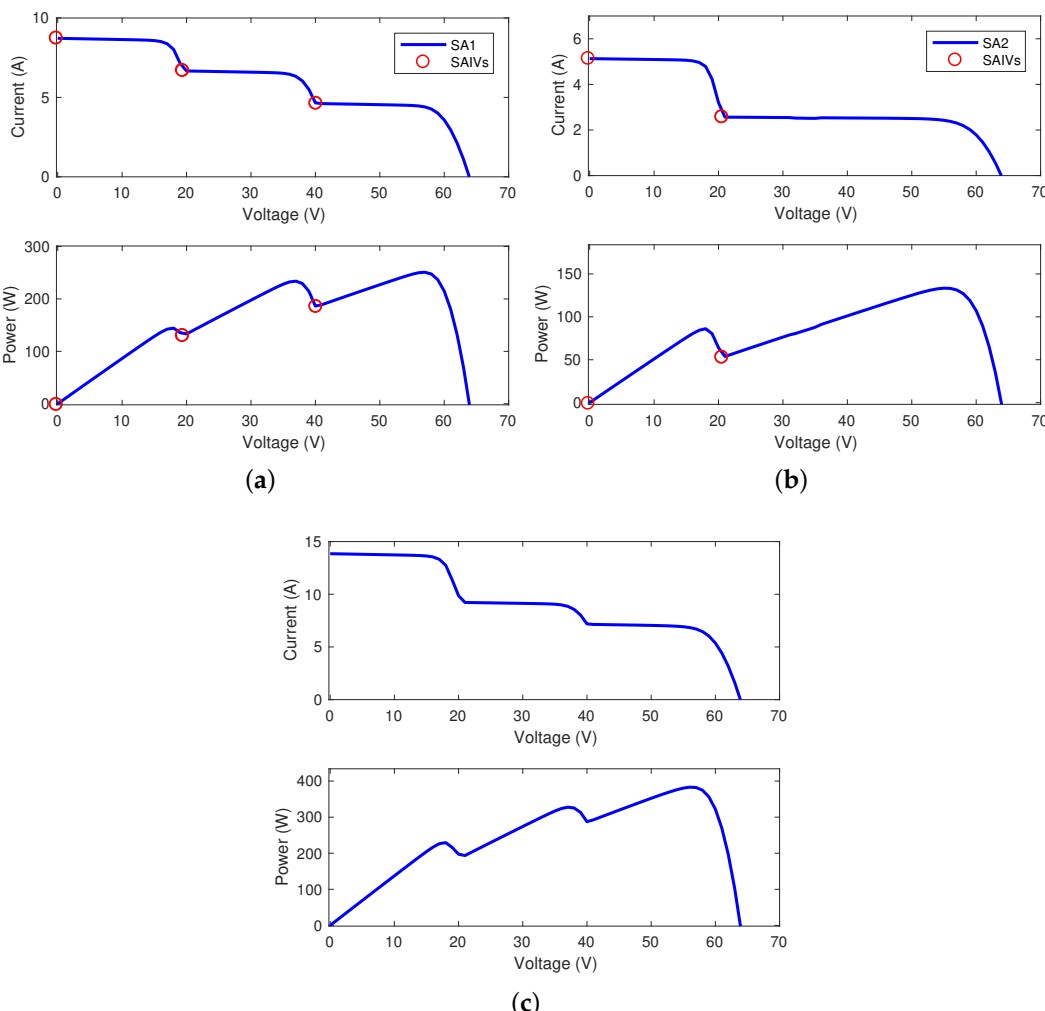

**Figure 9.** I-V curves for the calculation example (**a**) $SA1$, (**b**) $SA2$, and (**c**) array.

## 6. Performance Evaluation

In this section, the proposed model is compared with the model that was introduced in [14] (from here on reference model) in terms of the errors in the prediction of the array current and the computation time that is required to generate I-V curves. For all of the analyses, the ERDM 85SM/5 PV panel is used to generate the arrays; hence, the same SDM parameters that are defined in Section 4.4 are considered for the simulations.

## 6.1. Errors in the Current Prediction

In this analysis, two arrays are considered to evaluate the current error of the proposed and reference models with respect to the circuital implementation of the arrays in Matlab/Simulink (from here on circuital model). The proposed and reference models are both programmed in Matlab and solved with the function "fmincon" configured with "sqp-legacy" algorithm to provide a fair comparison. For the reference model, the lower and upper bounds for each node voltage are defined as $-1$ [V] and the array open-circuit voltage ($V_{oc,arr}$), respectively; while, for the proposed model, the lower and upper bounds of the node voltages are defined, as shown in Section 5 from the inflection voltages calculated, as explained in Section 4.

The two arrays are formed by three columns and two sub-arrays, as in the calculation example of Sections 4 and 5, where the first two columns form the $SA1$ and the last column forms the $SA2$. The matrix $M_{conn}$ for all of the arrays is formed by two columns, in the first one the odd elements are 1 and the even elements are 0, while, in the second column, all of the elements are 0. For example, the first column of $M_{conn}$ for the $5 \times 3$ array is $[0\ 1\ 0\ 1]^T$ and the second columns is $[0\ 0\ 0\ 0]^T$. Moreover, the matrix $M_{Iph}$ for the two arrays is formed by three equal vectors vectors: $I_{ph} \cdot [1\ 0.75\ 0.75\ 0.5\ 0.5]^T$ for the $5 \times 3$ array, and $I_{ph} \cdot [1\ 1\ 1\ 0.75\ 0.75\ 0.75\ 0.5\ 0.5\ 0.5\ 0.5]^T$ for the $10 \times 3$ array ($I_{ph} = 5.13$ [A]).

Figure 10 shows the I-V curves for the two evaluated arrays and Figure 11 introduces the errors in the current estimation of the proposed and reference models, regarding the circuital model. In Figure 10a, it can be observed that the proposed and reference models provide the same I-V curve for the $5 \times 3$ array, which is reflected in the same current errors, as illustrated in Figure 11a and the same value of the normalized sum of squared errors (NSSE) obtained for both models (0.0015%). For the $10 \times 3$ array, the I-V curves that are obtained with the proposed and reference models are almost the same (see Figure 10b); however, the reference model has significant errors around the second knee (between 105 [V] and 115 [V]), as shown in Figure 11b. Those errors can be evidenced in the NSSEs that were calculated for both I-V curves, which correspond to 0.0012% and 0.0027% for the proposed and reference models, respectively. Additionally, it is important to highlight that the proposed model provides similar NSSE values for the two arrays, demonstrating that it fits the circuital model.

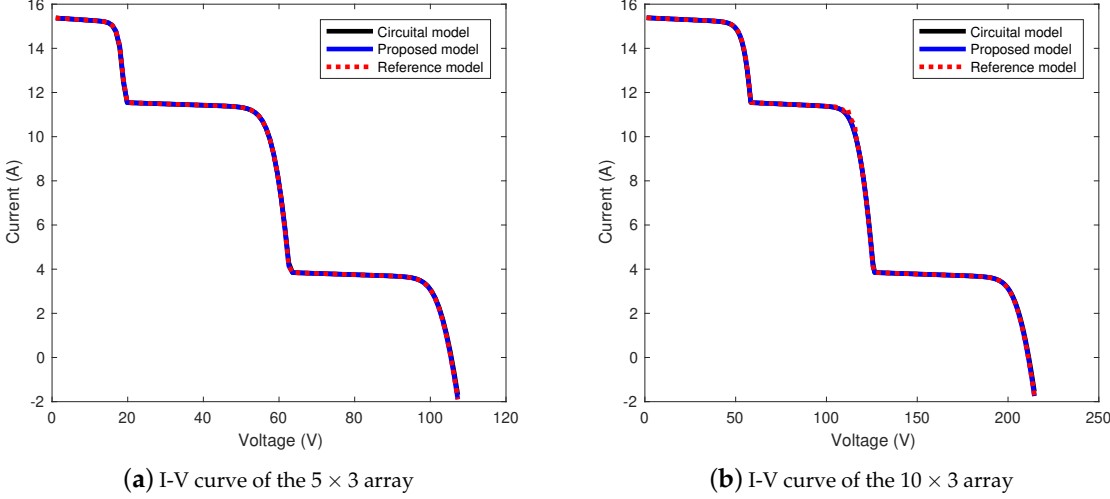

**(a)** I-V curve of the $5 \times 3$ array      **(b)** I-V curve of the $10 \times 3$ array

**Figure 10.** I-V curves of irregular arrays with different sizes with Circuital model (black line), Proposed model (blue line), and Reference model (dashed red line).

### 6.2. Calculation Time for Different Number of Rows

The proposed and reference models are simulated for arrays formed by two sub-arrays and increasing the number of rows from three to fifteen in order to evaluate their calculation times. The structure of the arrays is the same used in the previous section; therefore, the matrices $M_{conn}$ and $M_{Iph}$ are generated as explained before. The I-V curve of each array is calculated ten times in order to calculate its average value and standard deviation, which are introduced in Table 1 and shown graphically in Figure 12.

From the calculation times that are shown in Table 1 and Figure 12, it can be observed that the reference model is faster than the proposed model for a small array (three rows). This behavior can is expected due to the additional calculations that are required to obtain the inflection voltages. However, for arrays with four rows or more, the calculation time of the proposed model is less than the one of the reference model. Moreover, Figure 12 shows that the calculation time of the proposed model is approximately linear regarding the number rows; while, the calculation time of the reference model has a significant increment in the slope for arrays with more than nine rows. Hence, the results in Table 1 and Figure 12 illustrate the reduction in the calculation time that was obtained by defining the upper and lower bounds of the sub-arrays node voltages from the calculation of the inflection voltages.

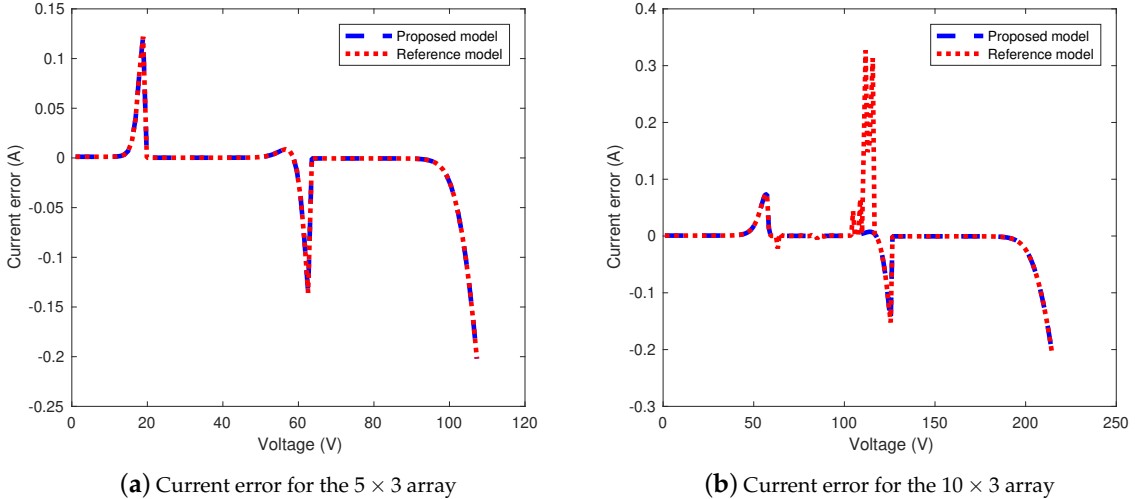

(**a**) Current error for the $5 \times 3$ array　　　　　　　(**b**) Current error for the $10 \times 3$ array

**Figure 11.** Current errors for irregular arrays with different sizes with Proposed model (dashed blue line) and Reference model (dotted red line) regarding Circuital model.

**Table 1.** The calculation time of the proposed and reference models for an array with three columns and rows from three to fifteen.

| Rows | Proposed Model | | Reference Model | |
|---|---|---|---|---|
| | Average (s) | Std (s) | Average (s) | Std (s) |
| 3 | 6.396 | 0.509 | 5.362 | 0.169 |
| 4 | 8.462 | 0.079 | 8.934 | 0.166 |
| 5 | 10.893 | 0.463 | 12.190 | 0.157 |
| 6 | 13.616 | 0.336 | 15.160 | 0.296 |
| 7 | 16.413 | 0.180 | 19.735 | 0.199 |
| 8 | 18.025 | 0.792 | 25.353 | 0.767 |
| 9 | 20.232 | 0.229 | 31.533 | 0.357 |
| 10 | 24.212 | 0.307 | 46.348 | 0.456 |
| 11 | 27.771 | 0.121 | 59.698 | 0.352 |
| 12 | 34.096 | 0.294 | 77.808 | 2.059 |
| 13 | 38.008 | 0.310 | 89.932 | 1.159 |
| 14 | 42.471 | 0.224 | 100.773 | 0.531 |
| 15 | 46.995 | 0.214 | 122.431 | 0.555 |

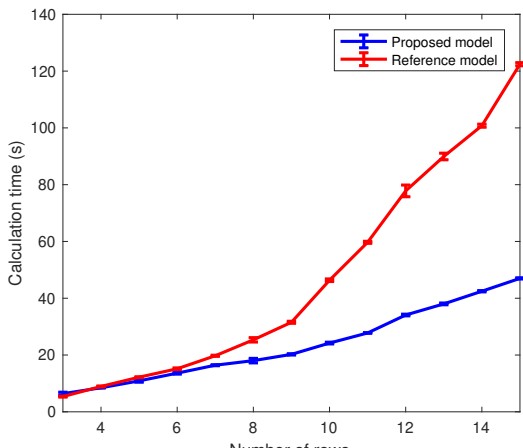

**Figure 12.** Calculation time of proposed and reference model for an array with three columns and rows from three to fifteen.

## 7. Application Example: Reconfiguration of Pv Arrays

In this section, the proposed model is used for a basic model-based reconfiguration PV array in order to illustrate its usefulness to model arrays in different configurations and the advantages of the reduction in the calculation time. The array that is considered in this section is formed by ERDM 85SM/5 modules; then, the SDM parameters that are defined in Section 4.4 are used.

Figure 13 illustrates the array structure, which is formed by three columns of fourteen modules each. Twelve modules are reconfigurable through the five switches that are shown in Figure 13, which are controlled by the reconfiguration algorithm. The rest of the array is fixed and the connections between the columns are also illustrated in Figure 13. This strategy is used to reconfigure only the array section that is subjected to partial shading without introducing reconfigurable switches that will not be used. A previous example of this strategy was reported in [26].

For the example, it is assumed that the reconfiguration system evaluates all of the possible arrays that can be implemented with the five switches to find the best configuration. Hence, considering that there are

five switches, there are $2^5$ possible configurations to evaluate. For each possibility, the reconfiguration system calculates the I-V and P-V curves (with voltage steps of 1 [V]) using a model to identify the global maximum power point (GMPP). Moreover, those calculations should be performed as fast as possible due to the shading profiles over the array change with the time due to the paths of the sun in the sky. Additionally, the time that is srequired to evaluate the 32 configurations determines the minimum update period of the reconfiguration system.

The reconfiguration system is evaluated for three shading profiles that are defined with the matrix $M_{Iph}$. Table 2 introduces the elements of $M_{Iph}$ for the reconfigurable part of the array, while the elements of $M_{Iph}$ for the fixed part of the array are defined as 2.5668 [A], i.e., for the first two columns $M_{Iph}(a,b) = 2.5668$ [A] $\forall$ $(a \in [7 \cdots 14] \wedge b \in [1 \cdots 2])$ and for the last column $M_{Iph}(a,b) = 2.5668$ [A] $\forall$ $(a \in [1 \cdots 14] \wedge b = 3)$.

**Table 2.** $I_{ph}$ currents (in [A]) of the reconfigurable modules of the array.

| Shading Profile 1 | | Shading Profile 2 | | Shading Profile 2 | |
|---|---|---|---|---|---|
| 5.1337 | 5.1337 | 5.1337 | 3.5936 | 5.1337 | 5.1337 |
| 3.5936 | 5.1337 | 3.5936 | 5.1337 | 3.5936 | 5.1337 |
| 3.5936 | 3.5936 | 1.5401 | 1.5401 | 3.5936 | 3.5936 |
| 1.5401 | 3.5936 | 5.1337 | 3.5936 | 1.5401 | 1.5401 |
| 1.5401 | 1.5401 | 1.5401 | 5.1337 | 1.5401 | 5.1337 |
| 5.1337 | 0.5134 | 5.1337 | 1.5401 | 5.1337 | 5.1337 |

The 32 possible configurations are evaluated while using the proposed and reference models for each shading profile to compare their calculation times and determine the minimum update period of the reconfiguration algorithm. Table 3 intorduces the calculation times for both models, which shows that the proposed model is between 1.8 and 2.1 times faster than the reference model. Such differences in the computation times results in a reconfiguration period of 20 [min] and 40 [min] with the proposed and reference models, respectively. Therefore, a reconfiguration system with the proposed model may reconfigure the array a double number of times with respect to a reconfiguration system with the reference model. Such an increment in the number of reconfigurations is translated into a better mitigation of the mismatching effects and an increment in the array power production.

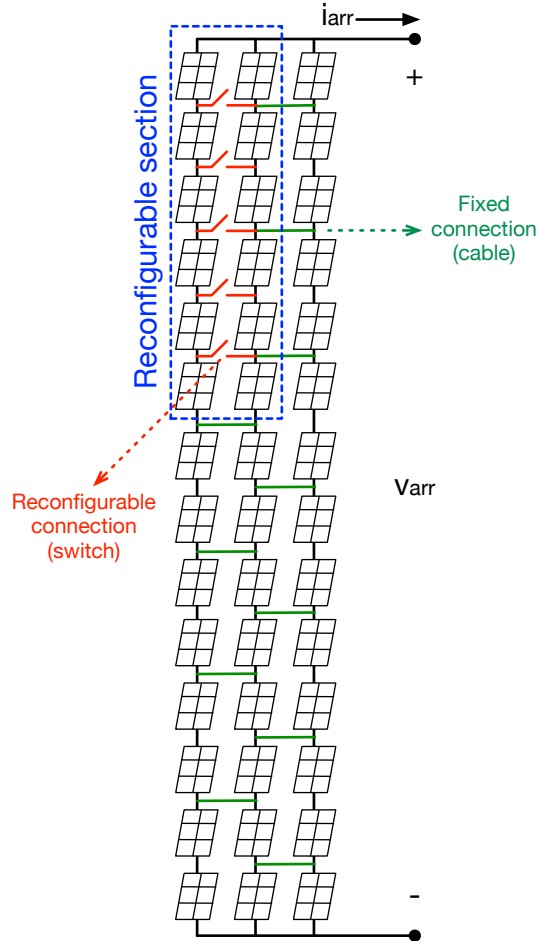

**Figure 13.** $14 \times 3$ array used for the reconfiguration example.

**Table 3.** Calculation time (in [min]) of the proposed and reference models to evaluate the 32 configurations for each shading profile.

| Shading Profile | Proposed Model | Reference Model |
|:---:|:---:|:---:|
| 1 | 18.2180 | 38.0085 |
| 2 | 19.1983 | 34.7788 |
| 3 | 17.9341 | 38.2907 |

## 8. Conclusions

A mathematical model for regular and irregular PV arrays, based on the inflection points concept, has been proposed. In this model, the array is divided into sub-arrays that are solved independently, which simplifies the model calculation. The main contribution of this model is the reduction of the processing time by using the inflection point concept to limit the search range of the node voltages in the solution of each sub-array, which enables analyzing larger arrays in the same time interval when compared with a previously published model.

Another contribution of the paper is the pseudocode that was developed to evaluate the system of nonlinear equations $(F_o(\vec{V}_n))$ for any regular or irregular configuration, which is used by a numerical method to calculate the sub-array inflection voltages for a given module in short-circuit. Moreover, the paper also provides an algorithm for calculating all of the sub-array inflection voltages, which are organized

into the $N_n + 1$ matrices ($M_{Vo}$ and $M_{Vno}^k \ \forall \ k \in [1 \cdots N_n]$), where $N_n$ is the number of nodes in the sub-array. From these matrices, the paper proposes a procedure for generating $N_n$ vectors ($V_{Vno}^k \ \forall \ k \in [1 \cdots N_n]$), which are used to define the upper and lower bounds of the solution of each sub-array node voltage. Those procedures are clearly illustrated through a calculation example of a small array that formed by two sub-arrays, which are useful for implementing the model in any programming language or platform.

The proposed model was evaluated by comparing its calculation time and estimation errors with the ones of the general model that was proposed in [14], which was defined as reference model. Both the proposed and reference model were solved while using the same numerical method to provide a fair comparison. Moreover, the estimation errors in the PV current of both methods were evaluated in contrast with the results provided by the circuital implementation of the same PV arrays using Simulink. In those tests, the proposed model reproduced the I-V curves of $5 \times 3$ and $10 \times 3$ arrays with a normalized sum of squared errors (NSSE) of 0.0015% and 0.0012%, respectively; while, the reference model obtained NSSE values of 0.0015% ($5 \times 3$) and 0.0027% ($10 \times 3$). Those NSSE results show that the proposed model reproduces, with higher accuracy, the larger PV arrays in comparison with the reference model. Such an improved convergence is achieved, in the proposed model, by reducing the search range of the node voltages by means of the inflection points calculation.

The calculation times of the proposed and reference models were evaluated with arrays that formed by three columns and a different number of rows. The calculation times of the proposed model were shorter for arrays with more than three rows, and those times grow linearly with the number of rows. Instead, the calculation times of the reference model grow exponentially with the number of rows, which put into evidence the improvement of the proposed solution over the reference general model: a faster and accurate model for estimating the power production of PV arrays under any electrical configuration and shading conditions.

The usefulness of the proposed model was also illustrated with a simple reconfiguration system that evaluates all of the possible configurations in order to determine the one with the highest power production. The reconfiguration system with the proposed model was approximately two times faster than the reference model for that particular case, but higher differences will appear for larger PV arrays. Such an increment in the calculation speed results in a reduction in the reconfiguration period and, consequently, an increment in the array power production along the day, since the best configuration is updated more frequently.

The only disadvantage of this new model, in comparison with the reference general model, concerns the additional memory that is required to store the matrices needed to calculate the inflection points. Therefore, a future development must be needed to reduce the memory requirements of the model, which could be focused on reallocating the memory that was used in the inflection points calculation to be used in the PV current calculation; hence, reaching the same memory requirements of the reference model. Another important future study concerns the implementation of the proposed model on embedded devices, which requires algorithms that are optimized for small platforms. Such a development is needed in order to deploy reconfiguration platforms that are based on the proposed fast model. Finally, the model is restricted to rectangular array configurations; therefore, all of the strings must have the same number of modules, which is a common practice in commercial PV installations. However, residential PV arrays could be formed by strings with a different number of modules due to roof space limitations; hence, a future study must consider the extension of the proposed model to support those kinds of PV arrays.

**Author Contributions:** Conceptualization, J.D.B.-R., L.A.T.-G. and C.A.R.-P.; methodology, J.D.B.-R. and C.A.R.-P.; software, J.D.B.-R. and L.A.T.-G.; validation, J.D.B.-R., L.A.T.-G. and C.A.R.-P.; formal analysis, J.D.B.-R., L.A.T.-G. and C.A.R.-P.; investigation, J.D.B-R., L.A.T.-G. and C.A.R.-P.; data curation, J.D.B.-R. and L.A.T.-G.; writing, review and editing, J.D.B.-R., L.A.T.-G. and C.A.R.-P.; visualization, J.D.B.-R., L.A.T.-G. and C.A.R.-P.; supervision, J.D.B.-R., L.A.T.-G. and C.A.R.-P. All authors have read and agreed to the published version of the manuscript.

**Funding:** The authors thank the Instituto Tecnológico Metropolitano for the APC payment. Moreover, this study was supported by the research group Materiales Avanzados y Energía of the Instituto Tecnológico Metropolitano and the Universidad Nacional de Colombia.

**Conflicts of Interest:** The authors declare no conflict of interest.

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
