# Peer review of "Mathematical Model for Regular and Irregular PV Arrays with Improved Calculation Speed"

_sustainability, doi:10.3390/su122410684_

Round 1

Reviewer 1 Report

The manuscript presents an interesting topic by proposing a modeling approach for estimating the power production of regular and irregular PV arrays under partial shading conditions. The authors claim that this model can provide a faster calculation in comparison with another general model proposed in the literature, but without reducing the prediction accuracy. The manuscript is well structured and written, and a good effort has been gone to make it easily readable and understandable. Additionally, the manuscript discussed one of the problematic issues facing power generation from the PV system under partial shading conditions.

Author Response

Dear Reviewer, thanks a lot for your kind comments.

Reviewer 2 Report

Dear authors,

Thank you for submitting your paper to the Sustainability Journal. .

Comments:

  1. It is not clear what the research gap that the paper is addressing. What is the objective of this paper? Please clarify somewhere clearly all your contributions.
  2. The literature review is not goal oriented. The process should be as follows:
  3. i) Critical evaluation of the literature; ii) identifying the gap based on this critical evaluation of the literature; iii) proposing your hypothesis to address the identified gap; iv) posing the appropriate and relevant research question based on your proposed hypothesis; and finally explaining your proposed method to answer this research question. Therefore, you will have a systematic way of conducting your research. Right now, the literature review section has no clear objective.
  4. An adequate literature review and a clear gap identification have been tried to be conducted. However, authors have ignored some research which has been done in the area. I strongly recommend the authors to provide a more comprehensive literature review in the introduction section. The following papers are recommended:
  • A method to detect photovoltaic array faults and partial shading in PV systems. IEEE Journal of Photovoltaics, 6(5), 1278-1285.
  • Impact of Partial Shading on Various PV Array Configurations and Different Modeling Approaches: A Comprehensive Review. IEEE Access, 8, 181375-181403.
  • Clean Development Mechanism, a bridge to mitigate the Greenhouse Gasses: is it broken in Iran. In 13th International Conference on Clean Energy (ICCE), Istanbul, pp 399-404.
  • A procedure for modeling photovoltaic arrays under any configuration and shading conditions. Energies, 11(4), 767.
  • Reconfiguration strategies for reducing partial shading effects in photovoltaic arrays: State of the art. Solar Energy, 182, 429-452.
  • Performance evaluation of complex electricity generation systems: A dynamic network-based data envelopment analysis approach. Energy Economics 91, 104894.
  • Simple and efficient approach to detect and diagnose electrical faults and partial shading in photovoltaic systems. Energy Conversion and Management, 196, 330-343.
  • Outlook on biofuels in future studies: A systematic literature review. Renewable and Sustainable Energy Reviews 134, 110326.
  • Adaptive GA-based reconfiguration of photovoltaic array combating partial shading conditions. Neural Computing and Applications, 30(4), 1145-1170.
  • Energy policy in Iran and international commitments for GHG emission reduction. Journal of environmental science and technology 17(1), 183-198.

  • Metaheuristic based comparative MPPT methods for photovoltaic technology under partial shading condition. Energy, 212, 118592.
  • A combined model of scenario planning and assumption-based planning for futurology, and robust decision making in the energy sector. Journal of energy policy and planning research 2 (2), 7-32.
  • A dynamic particles MPPT method for photovoltaic systems under partial shading conditions. Energy Conversion and Management, 220, 113070.
  1. A thorough editorial check and English improvement are needed. Please kindly proofread the entire manuscript.
  2. The conclusion part is also needed to be revised; which questions are answered, what is the value/originality/contribution of the paper, how the proposed method answers the research questions that previous methods are not able to answer?
  3. It feels you need a king of aggregating results somewhere clearer.
  4. Please propose and suggest more possible future studies related to the current study.
  5. If you can, please make a small comparison between what did you do and what others did before; as a conclusion.
  6. The abstract is not deep enough and Is not well prepared. Please try to re-write it better. The problem should be clearly stated and the gap which you are going to address need to be clarified. Simply explain your contributions and key findings.
  7. There are some errors in your reference list. Please check and fix the errors.

Author Response

Reviewer 2

Dear authors,
Thank you for submitting your paper to the Sustainability Journal.

Reply: Dear Reviewer, thanks a lot for your kind comments and useful suggestions.

Reviewer Point P 2.1 — It is not clear what the research gap that the paper is addressing. What is the objective of this paper? Please clarify somewhere clearly all your contributions.

Reply: The main contributions of the paper are:

  • A general model able to predict the power production of regular and irregular PV arrays with im-

    proved speed and convergence rate in comparison with other general models reported in literature.

  • The adoption of the inflection points concept to limit the search range of the node voltages, which enables to analyze larger arrays in the same time interval when compared with a previously published model.

  • The pseudocode developed to evaluate the system of nonlinear equations for any regular or irregular configuration, and the algorithm to calculate all the sub-array inflection voltages. Both procedures are clearly illustrated through a calculation example of a small array, which are useful to implement the model in any programming language or platform.

Following the reviewer suggestion, those contributions were clarified as follows:

In the Abstract: ”Photovoltaic (PV) systems are usually developed by configuring the PV arrays with regular connection schemes such as series-parallel, total cross-tied, bridge-linked, among others. Such a strategy is aimed at increasing the power generated by the PV system under partial shading conditions, since the power production changes depending on the connection scheme. Moreover, irregular and non-common connection schemes could provide higher power production for irregular (but realistic) shading conditions caused by threes or other objects. However, there are few mathematical models able to predict the power production of different configurations and reproduce the behavior of both regular and irregular PV arrays. Those general arrays models are slow due to the large amount of computations needed to find the PV current for a given PV voltage. Therefore, this paper proposes a general mathematical model to predict the power production of regular and irregular PV arrays, which provides a faster calculation in comparison with the general models reported in the literature, but without reducing the prediction accuracy. The proposed modeling approach is based on detecting the inflection points caused by the bypass diodes activation, which enables to narrow the range in which the modules voltages are searched, thus reducing the calculation time. Therefore, this fast model is useful to design the fixed connections of PV arrays subjected to shading conditions, to reconfigure the PV array in real- time depending on the shading pattern, among other applications. The proposed solution is validated by comparing the results with another general model and with a circuital implementation of the PV system.”

In the Introduction: ”The modeling procedures introduced in [14] and [15] allow modeling a PV array with any size and connected in any connection scheme. Both procedures use the single diode model to represent the modules in a PV panel, and they are based on the circuital nodes and meshes principles to define a system of nonlinear equations that represents the array. That system of equations is solved to finally obtain the global output current of the array for a given voltage; then, performing a voltage sweep it is possible to calculate the array I-V and P-V curves. However, the solutions of the models proposed in [14] and [15] require a high number of both mathematical operations and numerical method iterations due to the large search space (voltage and current range) in which the solution must be searched for; hence, the calculation times could be long. Therefore, that general solution could be impractical for some PV application such as: dynamic or static reconfiguration including heuristic methods [16], [17] in which the calculation of the power provided by the array must be done in short time; validation and evaluation of Maximum Power Point Tracking (MPP) techniques, as the ones introduced in [18] and [19], in which optimization algorithms are implemented in order to improve the performance of the control stage in PV systems; or even to provide an optimal design for large PV plants. With the procedure presented in [14] a 10×5 irregular PV array takes 10 minutes and 8 seconds to be solved for a given irradiance and temperature conditions, while by using the procedure introduced in [15] the time is reduced to 5 minutes and 18 seconds. Despite the improvement in the execution time, in a planning of PV systems or reconfiguration scenario, it could be impractical if several configurations must be evaluated in order to define the best for a given operating condition; moreover, the execution times will increase if a larger array is evaluated.

In the previous literature analysis two key points are identified. The first one is that there is a lack of fast modeling techniques able to analyze PV arrays in any configuration with any size, since the models proposed in [14,15] require high number of mathematical operations which imply high execution times. Those high execution times make those models impractical for important applications like reconfiguration, MPPT, or PV array sizing. The second key point is that the inflection points concept has been used to reduce the solution time of SP arrays models; however, the inflection points concept has not been used for models of other PV array configurations or for the general models proposed in [14,15].

Therefore, this paper is based on the following hypothesis: it is possible to improve the computation times for analyzing PV arrays with any size and configuration by combining the inflection points modeling technique of [10] with the circuital nodes principle presented in [14]. In this way, the new solution provides the same analytical versatility to model any PV array (any size and configuration) but with much shorter processing times. This feature will be useful for reconfiguration techniques, analysis and design of large PV fields, among others. Moreover, the adoption of the inflection points concept also improves the convergence rate of the general model, which reduced the prediction errors in comparison with the models reported in [14] and [15].”

In the Conclusions: ”A mathematical model for regular and irregular PV arrays, based on the inflection points concept, has been proposed. In this model, the array is divided into sub-arrays that are solved independently, which simplifies the model calculation. The main contribution of this model is the reduction of the processing time by using the inflection point concept to limit the search range of the node voltages in the solution of each sub-array, which enables to analyze larger arrays in the same time interval when compared with a previously published model.

Another contribution of the paper is the pseudocode developed to evaluate the system of nonlinear equations (Fo(Vn)) for any regular or irregular configuration, which is used by a numerical method to calculate the sub-array inflection voltages for a given module in short-circuit. Moreover, the paper also provides an algorithm to calculate all the sub-array inflection voltages, which are organized into the Nn+1matrices (MVo and MkVno ∀ k ∈[1···Nn]), where Nn is the number of nodes in the sub-array. From these matrices, the paper proposes a procedure to generate Nn vectors (VkVno ∀ k ∈ [1 · · · Nn]), which are used to define the upper and lower bounds of the solution of each sub-array node voltage. Those procedures are clearly illustrated through a calculation example of a small array formed by two sub-arrays, which are useful to implement the model in any programming language or platform.

The proposed model was evaluated by comparing its calculation time and estimation errors with the ones of the general model proposed in [14], which was defined as reference model. Both the proposed and reference model were solved using the same numerical method to provide a fair comparison. Moreover, the estimation errors in the PV current of both methods were evaluated in contrast with the results provided by the circuital implementation of the same PV arrays using Simulink. In those tests, the proposed model reproduced the I-V curves of 5 × 3 and 10 × 3 arrays with normalized sum of squared errors (NSSE) of 0.0015% and 0.0012%, respectively; while the reference model obtained NSSE values of 0.0015% (5×3) and 0.0027% (10×3). Those NSSE results show that the proposed model reproduces with higher accuracy the larger PV arrays in comparison with the reference model. Such an improved convergence is achieved, in the proposed model, by reducing the search range of the node voltages by means of the inflection points calculation.”

Reviewer Point P 2.2 — The literature review is not goal oriented. The process should be as follows: 3. i) Critical evaluation of the literature; ii) identifying the gap based on this critical evaluation of the literature; iii) proposing your hypothesis to address the identified gap; iv) posing the appropriate and relevant research question based on your proposed hypothesis; and finally explaining your proposed method to answer this research question. Therefore, you will have a systematic way of conducting your research. Right now, the literature review section has no clear objective.

Reply: We thank the reviewer for his comment. We have modified the introduction following the suggestions. We have introduced some paragraphs to clarify the gaps identified in the literature revision; in the same way, the contribution of our work was also clarified:

In lines 32 to 38: a description of a PV array is given to introduce the issue of its modeling.

A typical PV array is formed by connecting multiple modules in series to form strings and multiple strings in parallel to form the array (i.e. Series-Parallel configuration). Each module can be represented by an equivalent circuit; therefore, the model of a PV array is obtained by analyzing the array equivalent circuit obtained by the interconnection of the modules. If all the modules are exactly the same and operate under the same irradiance and temperature conditions, all the array can modeled by scaling the model of one module. Nonetheless, in real operating conditions the arrays are subjected to partial shading and the modules are different due to aging and manufacturing tolerances [3].

In lines 42 to 46: additional information concerning the different PV cells models founded in literature was included.

An interesting combination single-diode and double-diode models is proposed in [5]. In this work the authors use a machine learning-based technique to calculate the power of PV arrays using single and double-diode module models. Nevertheless, this solution does not apply to any configuration and it does not consider partial shading or mismatching operating conditions.

In lines 53 to 58: a better description of the modeling techniques reported in literature for different PV array configurations is given.

Different modeling procedures have been reported in the literature. Some of them apply only for SP [4,8-10] or TCT arrays [11,12]; while others [7,13] perform an independent analysis for SP, TCT, BL, and HC arrays, but they do not propose systematical procedure to model arrays in any configuration. Only in [14] and [15] the authors propose modeling procedures for arrays in any configuration. However, those works are focused on the generation of a system of nonlinear equations to model an array and the solution of the models requires high calculation times.

In lines 70 to 113: a more detailed analysis of previous works concerning PV array modeling was introduced. In this way, the identified researching gaps were clarified and the hypothesis of our work is given.

The inflection points concept was also used in [9] to reduce the calculation time for solving the model of SP arrays. In this paper the authors analyze each string independently and use the single-diode model to represent each module of the array. Then, each string is modeled by a nonlinear equation where the string current is the unknown variable. The authors use the inflection points to identify the active and inactive modules to reduce the complexity of the nonlinear equation of each string, thus reducing the calculation burden of the numerical method that solves the equation. Moreover, the inflection points are also used to restrict the search range for the solution of the string current, which considerably reduces the solution time.

The modeling procedures introduced in [14] and [15] allow modeling a PV array with any size and connected in any connection scheme. Both procedures use the single diode model to represent the modules in a PV panel, and they are based on the circuital nodes and meshes principles to define a system of nonlinear equations that represents the array. That system of equations is solved to finally obtain the global output current of the array for a given voltage; then, performing a voltage sweep it is possible to calculate the array I-V and P-V curves. However, the solutions of the models proposed in [14] and [15] require a high number of both mathematical operations and numerical method iterations due to the large search space (voltage and current range) in which the solution must be searched for; hence, the calculation times could be long. Therefore, that general solution could be impractical for some PV application such as: dynamic or static reconfiguration including heuristic methods [16], [17] in which the calculation of the power provided by the array must be done in short time; validation and evaluation of Maximum Power Point Tracking (MPP) techniques, as the ones introduced in [18] and [19], in which optimization algorithms are implemented in order to improve the performance of the control stage in PV systems; or even to provide an optimal design for large PV plants. With the procedure presented in [14] a 10 × 5 irregular PV array takes 10 minutes and 8 seconds to be solved for a given irradiance and temperature conditions, while by using the procedure introduced in [15] the time is reduced to 5 minutes and 18 seconds. Despite the improvement in the execution time, in a planning of PV systems or reconfiguration scenario, it could be impractical if several configurations must be evaluated in order to define the best for a given operating condition; moreover, the execution times will increase if a larger array is evaluated.

In the previous literature analysis two key points are identified. The first one is that there is a lack of fast modeling techniques able to analyze PV arrays in any configuration with any size, since the models proposed in [14,15] require high number of mathematical operations which imply high execution times. Those high execution times make those models impractical for important applications like reconfiguration, MPPT, or PV array sizing. The second key point is that the inflection points concept has been used to reduce the solution time of SP arrays models; however, the inflection points concept has not been used for models of other PV array configurations or for the general models proposed in [14,15].

Therefore, this paper is based on the following hypothesis: it is possible to improve the computation times for analyzing PV arrays with any size and configuration by combining the inflection points modeling technique of [10] with the circuital nodes principle presented in [14]. In this way, the new solution provides the same analytical versatility to model any PV array (any size and configuration) but with much shorter processing times. This feature will be useful for reconfiguration techniques, analysis and design of large PV fields, among others. Moreover, the adoption of the inflection points concept also improves the convergence rate of the general model, which reduced the prediction errors in comparison with the models reported in [14] and [15].

Reviewer Point P 2.3 — An adequate literature review and a clear gap identification have been tried to be conducted. However, authors have ignored some research which has been done in the area. I strongly recommend the authors to provide a more comprehensive literature review in the introduction section. The following papers are recommended:

  • A method to detect photovoltaic array faults and partial shading in PV systems. IEEE Journal of Photovoltaics, 6(5), 1278-1285.

  • Impact of Partial Shading on Various PV Array Configurations and Different Modeling Approaches: A Comprehensive Review. IEEE Access, 8, 181375-181403.

  • Clean Development Mechanism, a bridge to mitigate the Greenhouse Gasses: is it broken in Iran. In 13th International Conference on Clean Energy (ICCE), Istanbul, pp 399-404.

  • A procedure for modeling photovoltaic arrays under any configuration and shading conditions. Energies, 11(4), 767.

  • Reconfiguration strategies for reducing partial shading effects in photovoltaic arrays: State of the art. Solar Energy, 182, 429-452.

  • Performance evaluation of complex electricity generation systems: A dynamic network-based data envelopment analysis approach. Energy Economics 91, 104894.

  • Simple and efficient approach to detect and diagnose electrical faults and partial shading in photovoltaic systems. Energy Conversion and Management, 196, 330-343.

  • Outlook on biofuels in future studies: A systematic literature review. Renewable and Sustainable Energy Reviews 134, 110326.

  • Adaptive GA-based reconfiguration of photovoltaic array combating partial shading conditions. Neural Computing and Applications, 30(4), 1145-1170.

  • Energy policy in Iran and international commitments for GHG emission reduction. Journal of environmental science and technology 17(1), 183-198.

  • Metaheuristic based comparative MPPT methods for photovoltaic technology under partial shading condition. Energy, 212, 118592.

  • A combined model of scenario planning and assumption-based planning for futurology, and robust decision making in the energy sector. Journal of energy policy and planning research 2 (2), 7-32.

  • A dynamic particles MPPT method for photovoltaic systems under partial shading conditions. Energy Conversion and Management, 220, 113070.

Reply: Following the reviewer suggestion, we considered the recommended papers. The papers included in the new version of our work are:

  • A procedure for modeling photovoltaic arrays under any configuration and shading conditions. Energies, 11(4), 767.

  • Impact of Partial Shading on Various PV Array Configurations and Different Modeling Approaches: A Comprehensive Review. IEEE Access, 8, 181375-181403.

  • Reconfiguration strategies for reducing partial shading effects in photovoltaic arrays: State of the art. Solar Energy, 182, 429-452.

  • Adaptive GA-based reconfiguration of photovoltaic array combating partial shading conditions. Neural Computing and Applications, 30(4), 1145-1170.

  • A dynamic particles MPPT method for photovoltaic systems under partial shading conditions. Energy Conversion and Management, 220, 113070.

  • Metaheuristic based comparative MPPT methods for photovoltaic technology under partial shad- ing condition. Energy, 212, 118592.

The aforementioned papers were included in the introduction and Section 2 to clarify the context of the research problem as follows:

In the Introduction: ”Therefore, that general solutions could be impractical for some PV applications such as dynamic or static reconfiguration including heuristic methods [16], [17] in which the calculation of the power provided by the array must be done in short time; validation and evaluation of Maximum Power Point Tracking (MPP) techniques as the introduced in [18] and [19] in which optimization algorithms are implemented in order to improve the performance of the control stage in PV systems; or even to provide an optimal design for large PV plants. With the procedure presented in [14] a 10 × 5 irregular PV array takes 10 minutes and 8 seconds to be solved for a given irradiance and temperature conditions, while by using the procedure introduced in [15] the time is reduced to 5 minutes and 18 seconds. Despite the improvement in the execution time, in a planning of PV systems or reconfiguration scenario, it could be impractical if several configurations must be evaluated in order to define the best for a given operating condition; moreover, the execution times will increase if a larger array is evaluated.

In the previous literature analysis two key points are identified. The first one is that there is a lack of fast modeling techniques able to analyze PV arrays in any configuration with any size, since the models proposed in [14,15] require high number of mathematical operations which imply high execution times. Those high execution times make those models impractical for important applications like reconfiguration, MPPT, or PV array sizing. The second key point is that the inflection points concept has been used to reduce the solution time of SP arrays models; however, the inflection points concept has not been used for models of other PV array configurations or for the general models proposed in [14,15].

Therefore, this paper is based on the following hypothesis: it is possible to improve the computation times for analyzing PV arrays with any size and configuration by combining the inflection points modeling technique of [10] with the circuital nodes principle presented in [14]. In this way, the new solution provides the same analytical versatility to model any PV array (any size and configuration) but with much shorter processing times. This feature will be useful for reconfiguration techniques, analysis and design of large PV fields, among others. Moreover, the adoption of the inflection points concept also improves the convergence rate of the general model, which reduced the prediction errors in comparison with the models reported in [14] and [15].”

In Section 2: ”However, according to [24] for combinations of different shading cases such short and long wide at irradiance levels from 200 W/m2 to 900 W/m2, Su-Do-Ku configuration exhibits the best performance compared to SP and TCT.”

The following papers address a very important issue in PV arrays operation such as detecting faults. The authors proposed procedures based on measuring certain variables (voltage, current, temperature and irradiance) which are processed to define if the system exhibit a fault. However, authors do not use a modeling process of the array to obtain the I-V and P-V curves. Therefore, we respectfully consider the papers do not contribute significantly in our work.

  • A method to detect photovoltaic array faults and partial shading in PV systems. IEEE Journal of Photovoltaics, 6(5), 1278-1285.

  • Simple and efficient approach to detect and diagnose electrical faults and partial shading in photovoltaic systems. Energy Conversion and Management, 196, 330-343.

The following papers address issues related with energy, renewable sources and electric systems. However, PV array modeling is not the main topic. Therefore, we respectfully think the suggested papers do not contribute significantly in our work.

  • Clean Development Mechanism, a bridge to mitigate the Greenhouse Gasses: is it broken in Iran. In 13th International Conference on Clean Energy (ICCE), Istanbul, pp 399-404.

  • Outlook on biofuels in future studies: A systematic literature review. Renewable and Sustainable Energy Reviews 134, 110326.

  • Performance evaluation of complex electricity generation systems: A dynamic network-based data envelopment analysis approach. Energy Economics 91, 104894.

    The following papers are written in arabic. We are not able to translate them. In addition, from the titles we consider the main topic of such papers is not PV array modeling. Therefore, we respectfully think the suggested papers do not contribute significantly in our work.

  • A combined model of scenario planning and assumption-based planning for futurology, and robust decision making in the energy sector. Journal of energy policy and planning research 2 (2), 7-32.

  • Energy policy in Iran and international commitments for GHG emission reduction. Journal of environmental science and technology 17(1), 183-198.

Reviewer Point P 2.4 — A thorough editorial check and English improvement are needed. Please kindly proofread the entire manuscript.

Reply: We thank the reviewer for his comment to improve the paper. We proofread all the manuscript and modified some sentences to make the ideas are clearer. Moreover, we corrected some typos along the paper.

Reviewer Point P 2.5 — The conclusion part is also needed to be revised; which questions are answered, what is the value/originality/contribution of the paper, how the proposed method answers the research questions that previous methods are not able to answer?

 Reply: Following the reviewer suggestion, the conclusions section was modified to highlight the paper contributions and improvements over previous works:

”A mathematical model for regular and irregular PV arrays, based on the inflection points concept, has been proposed. In this model, the array is divided into sub-arrays that are solved independently, which simplifies the model calculation. The main contribution of this model is the reduction of the processing time by using the inflection point concept to limit the search range of the node voltages in the solution of each sub-array, which enables to analyze larger arrays in the same time interval when compared with a previously published model.

Another contribution of the paper is the pseudocode developed to evaluate the system of nonlinear equations (Fo(Vn)) for any regular or irregular configuration, which is used by a numerical method to calculate the sub-array inflection voltages for a given module in short-circuit. Moreover, the paper also provides an algorithm to calculate all the sub-array inflection voltages, which are organized into the

Nn+1 matrices (MVo and MkVno ∀ k ∈[1···Nn]), where Nn is the number of nodes in the sub-array. From these matrices, the paper proposes a procedure to generate Nn vectors (VkVno ∀ k ∈ [1 · · · Nn]), which are used to define the upper and lower bounds of the solution of each sub-array node voltage. Those procedures are clearly illustrated through a calculation example of a small array formed by two sub-arrays, which are useful to implement the model in any programming language or platform.

The proposed model was evaluated by comparing its calculation time and estimation errors with the ones of the general model proposed in [14], which was defined as reference model. Both the proposed and reference model were solved using the same numerical method to provide a fair comparison. Moreover, the estimation errors in the PV current of both methods were evaluated in contrast with the results provided by the circuital implementation of the same PV arrays using Simulink. In those tests, the proposed model reproduced the I-V curves of 5 × 3 and 10 × 3 arrays with normalized sum of squared errors (NSSE) of 0.0015% and 0.0012%, respectively; while the reference model obtained NSSE values of 0.0015% (5×3) and 0.0027% (10×3). Those NSSE results show that the proposed model reproduces with higher accuracy the larger PV arrays in comparison with the reference model. Such an improved convergence is achieved, in the proposed model, by reducing the search range of the node voltages by means of the inflection points calculation.

The calculation times of the proposed and reference models were evaluated with arrays formed by three columns and different number of rows. The calculation times of the proposed model were shorter for arrays with more than three rows, and those times grow linearly with the number of rows. Instead, the calculation times of the reference model grow exponentially with the number of rows, which put into evidence the improvement of the proposed solution over the reference general model: a faster and accurate model for estimating the power production of PV arrays under any electrical configuration and shading conditions.

The usefulness of the proposed model was also illustrated with a simple reconfiguration system that evaluates all the possible configurations to determine the one with the highest power production. The reconfiguration system with the proposed model was approximately two times faster than the reference model for that particular case, but higher differences will appear for larger PV arrays. Such an increment in the calculation speed results in a reduction in the reconfiguration period and, as consequence, an increment in the array power production along the day since the best configuration is updated more frequently.

The only disadvantage of this new model, in comparison with the reference general model, concerns the additional memory required to store the matrices needed to calculate the inflection points. Therefore, a future development must be needed to reduce the memory requirements of the model, which could be focused on reallocating the memory used in the inflection points calculation to be used in the PV current calculation; hence, reaching the same memory requirements of the reference model. Another important future study concerns the implementation of the proposed model on embedded devices, which requires algorithms optimized for small platforms. Such a development is needed to deploy reconfiguration platforms based on the proposed fast model. Finally, the model is restricted to rectangular array configurations; therefore, all the strings must have the same number of modules, which is a common practice in commercial PV installations. However, residential PV arrays could be formed by strings with different number of modules due to roof space limitations; hence, a future study must consider the extension of the proposed model to support those kind of PV arrays.”

Reviewer Point P 2.6 — It feels you need a king of aggregating results somewhere clearer.

 Reply: We have summarized the numerical results in the conclusions section, which provides a clear measurement of the performance provided by the proposed model. In this way, the reader will find an aggregated discussion of the contribution and performance of the new solution:

”The proposed model was evaluated by comparing its calculation time and estimation errors with the ones of the general model proposed in [14], which was defined as reference model. Both the proposed and reference model were solved using the same numerical method to provide a fair comparison. Moreover, the estimation errors in the PV current of both methods were evaluated in contrast with the results provided by the circuital implementation of the same PV arrays using Simulink. In those tests, the proposed model reproduced the I-V curves of 5 × 3 and 10 × 3 arrays with normalized sum of squared errors (NSSE) of 0.0015% and 0.0012%, respectively; while the reference model obtained NSSE values of 0.0015% (5×3) and 0.0027% (10×3). Those NSSE results show that the proposed model reproduces with higher accuracy the larger PV arrays in comparison with the reference model. Such an improved convergence is achieved, in the proposed model, by reducing the search range of the node voltages by means of the inflection points calculation.

The calculation times of the proposed and reference models were evaluated with arrays formed by three columns and different number of rows. The calculation times of the proposed model were shorter for arrays with more than three rows, and those times grow linearly with the number of rows. Instead, the calculation times of the reference model grow exponentially with the number of rows, which put into evidence the improvement of the proposed solution over the reference general model: a faster and accurate model for estimating the power production of PV arrays under any electrical configuration and shading conditions. ”

Reviewer Point P 2.7 — Please propose and suggest more possible future studies related to the current study.

Reply: Following the reviewer suggestion, three additional future studies related to the proposed model have been included in the last paragraph of the conclusions section:

”The only disadvantage of this new model, in comparison with the reference general model, concerns the additional memory required to store the matrices needed to calculate the inflection points. Therefore, a future development must be needed to reduce the memory requirements of the model, which could be focused on reallocating the memory used in the inflection points calculation to be used in the PV current calculation; hence, reaching the same memory requirements of the reference model. Another important future study concerns the implementation of the proposed model on embedded devices, which requires algorithms optimized for small platforms. Such a development is needed to deploy reconfiguration platforms based on the proposed fast model. Finally, the model is restricted to rectangular array configurations; therefore, all the strings must have the same number of modules, which is a common practice in commercial PV installations. However, residential PV arrays could be formed by strings with different number of modules due to roof space limitations; hence, a future study must consider the extension of the proposed model to support those kind of PV arrays.”

Reviewer Point P 2.8 — If you can, please make a small comparison between what did you do and what others did before; as a conclusion.

Reply: The proposed model was evaluated by comparing its calculation time and estimation errors with the ones of the general model proposed in [14], which has been defined as the reference solution since it

also provides a general model for any array configuration. Moreover, the main differences, advantages and disadvantages between the proposed and reference models have been clarified.

Therefore, following the reviewer suggestions, the following modifications have been performed in the conclusions section:

First, the main contributions of the proposed solution, over a previously published model, have been described: ”A mathematical model for regular and irregular PV arrays, based on the inflection points concept, has been proposed. In this model, the array is divided into sub-arrays that are solved independently, which simplifies the model calculation. The main contribution of this model is the reduction of the processing time by using the inflection point concept to limit the search range of the node voltages in the solution of each sub-array, which enables to analyze larger arrays in the same time interval when compared with a previously published model.”

Second, the performance of both proposed and reference models was contrasted using numerical results, which put into evidence the improvements of the new solution: ”The proposed model was evaluated by comparing its calculation time and estimation errors with the ones of the general model proposed in [14], which was defined as reference model. Both the proposed and reference model were solved using the same numerical method to provide a fair comparison. Moreover, the estimation errors in the PV current of both methods were evaluated in contrast with the results provided by the circuital implementation of the same PV arrays using Simulink. In those tests, the proposed model reproduced the I-V curves of 5 × 3 and 10 × 3 arrays with normalized sum of squared errors (NSSE) of 0.0015% and 0.0012%, respectively; while the reference model obtained NSSE values of 0.0015% (5 × 3) and 0.0027% (10 × 3). Those NSSE results show that the proposed model reproduces with higher accuracy the larger PV arrays in comparison with the reference model. Such an improved convergence is achieved, in the proposed model, by reducing the search range of the node voltages by means of the inflection points calculation.

The calculation times of the proposed and reference models were evaluated with arrays formed by three columns and different number of rows. The calculation times of the proposed model were shorter for arrays with more than three rows, and those times grow linearly with the number of rows. Instead, the calculation times of the reference model grow exponentially with the number of rows, which put into evidence the improvement of the proposed solution over the reference general model: a faster and accurate model for estimating the power production of PV arrays under any electrical configuration and shading conditions.

The usefulness of the proposed model was also illustrated with a simple reconfiguration system that evaluates all the possible configurations to determine the one with the highest power production. The reconfiguration system with the proposed model was approximately two times faster than the reference model for that particular case, but higher differences will appear for larger PV arrays. Such an increment in the calculation speed results in a reduction in the reconfiguration period and, as consequence, an increment in the array power production along the day since the best configuration is updated more frequently.”

Moreover, Sections 6 and 7 describe in detail those comparisons. 

Finally, the main disadvantage of the proposed model, in comparison with the reference general model, was also discussed, which leads to propose a future study to improve the solution: ”The only disadvantage of this new model, in comparison with the reference general model, concerns the additional memory required to store the matrices needed to calculate the inflection points. Therefore, a future development must be needed to reduce the memory requirements of the model, which could be focused on reallocating the memory used in the inflection points calculation to be used in the PV current calculation; hence, reaching the same memory requirements of the reference model.”

Reviewer Point P 2.9 — The abstract is not deep enough and Is not well prepared. Please try to re-write it better. The problem should be clearly stated and the gap which you are going to address need to be clarified. Simply explain your contributions and key findings.

Reply: Following the reviewer suggestion, the abstract was modified to clarify the intended objective of the paper (contribution): the main objetive is to provide a fast mathematical model able to predict the power production of a PV array with any configuration, which reduced the calculation time in comparison with other general models already published in literature. Moreover, some applications that benefit from such a fast calculation are also briefly described.

Moreover, the main novelty (contribution) of the proposed solution is also clarified: use the inflection points concepto to narrow the search space of the model solutions, which reduced the calculation time (intended objective).

The modified Abstract is the following one:

”Photovoltaic (PV) systems are usually developed by configuring the PV arrays with regular connection schemes such as series-parallel, total cross-tied, bridge-linked, among others. Such a strategy is aimed at increasing the power generated by the PV system under partial shading conditions, since the power production changes depending on the connection scheme. Moreover, irregular and non-common connection schemes could provide higher power production for irregular (but realistic) shading conditions caused by threes or other objects. However, there are few mathematical models able to predict the power production of different configurations and reproduce the behavior of both regular and irregular PV arrays. Those general arrays models are slow due to the large amount of computations needed to find the PV current for a given PV voltage. Therefore, this paper proposes a general mathematical model to predict the power production of regular and irregular PV arrays, which provides a faster calculation in comparison with the general models reported in the literature, but without reducing the prediction accuracy. The proposed modeling approach is based on detecting the inflection points caused by the bypass diodes activation, which enables to narrow the range in which the modules voltages are searched, thus reducing the calculation time. Therefore, this fast model is useful to design the fixed connections of PV arrays subjected to shading conditions, to reconfigure the PV array in real-time depending on the shading pattern, among other applications. The proposed solution is validated by comparing the results with another general model and with a circuital implementation of the PV system.”

Reviewer Point P 2.10 — There are some errors in your reference list. Please check and fix the errors.

 Reply: According to the reviewer suggestion we reviewed all the references and corrected some errors. We found some missing information in references 1 and 2, and an error in the name of one of the authors in reference 12. Moreover, we eliminated the doi number from some references to avoid extra information in the reference section.

Round 2

Reviewer 2 Report

Dear authors,
Thank you for your efforts in revising the manuscript. I believe you did a great job in the revision and the changes alleviated my concerns regarding the manuscript. Therefore, I recommend its publication.
Looking forward to seeing the published version of your paper.
Good luck